# GaLore+: Boosting Low-Rank Adaptation for LLMs with Cross-Head Projection

## Abstract

Recent low-rank training methods, such as GaLore, have significantly reduced the memory required to optimize large language models (LLMs). However, these methods often suffer from time-consuming low-rank projection estimations. In particular, the singular value decomposition (SVD) in GaLore can consume more than 80% of the total training time. To address this issue, we propose GaLore+, which uses cross-head low-rank projection to reduce the substantial time consumption in estimating low-rank projections for multi-head attention. In addition, we employ randomized subspace iteration to achieve fast SVD. To further enhance performance, we propose sparsely coded residuals to reduce the errors caused by low-rank approximation on the first- and second-order moments of the optimizers and weight updates. We evaluate GaLore+ on arithmetic reasoning and natural language generation datasets. Our experiments demonstrate that GaLore+ delivers superior performance while achieving approximately $4\times$ fine-tuning speed compared to vanilla GaLore.

## 1 Introduction

As the sizes of language models grow rapidly, training models from scratch for different tasks becomes impractical due to the significant time and computational resources required. To address this challenge, current research and applications typically rely on pre-training large language models (LLMs) and subsequently fine-tuning them for specific downstream tasks. Such a paradigm has been proven highly efficient in a wide range of tasks, including, but not limited to, natural language processing, question-answering, and reasoning (Roziere et al., 2023; Li et al., 2023; Ouyang et al., 2022; Brown, 2020).

However, full fine-tuning of entire LLMs requires enormous memory, making it prohibitively expensive for individuals and start-ups. Parameter-efficient fine-tuning (PEFT) methods only fine-tune a small number of the weights, significantly reducing the memory requirements (Lialin et al., 2023; Ding et al., 2023). Among them, the methods based on low-rank reparameterization, such as LoRA (Hu et al., 2022), have attracted much attention due to their impressive efficiency. Recently, a low-rank adaptation method named GaLore (Zhao et al., 2024) demonstrated the ability to optimize an LLM with 7 billion parameters on a consumer-level GPU with 24 GB of memory. Low-rank adaptation methods achieve dramatic memory reduction by updating the

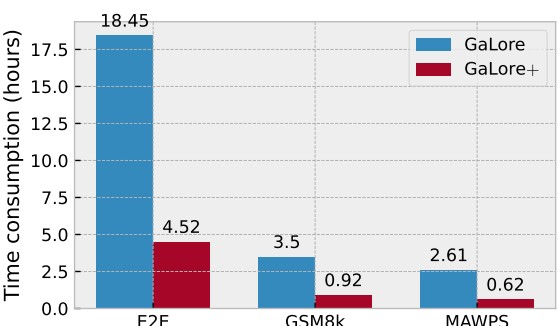

Figure 1: We compare the time consumption for fine-tuning LLaMA2-7B on different datasets with GaLore and GaLore+.

weights of LLMs in low-rank subspace, thus reducing the number of fine-tuned weights. We categorize existing low-rank adaptation methods into three groups based on how they construct low-rank projections *i.e.*, parameterized projection, random projection, and analytic projection.

**Algorithm 1** GaLore+ (PyTorch-like pseudocode)

```
1: for weight in model.parameters():
2:     grad = weight.grad
3:     # original space -> compact space
4:     # cross-head low-rank projection
5:     lor_grad, lor_proj = project(grad)              ▷ Section 4.1
6:     # update by AdamW or other optimizers
7:     lor_update, lor_moments = update(lor_grad)
8:     # sparsely coded residual
9:     res_update = estimate(grad, lor_proj, lor_moments)   ▷ Section 4.2
10:    # compact space -> original space
11:    update = project_back(lor_update) + res_update
12:    weight.data += update
```
*Note: The* green *background highlights the improvements over GaLore (Zhao et al., 2024).*

*i) Parameterized projection.* The representative work is LoRA (Hu et al., 2022), which approximates the parameter updates as the product of two trainable low-rank matrices. PiSSA (Meng et al., 2024) extends this by employing singular value decomposition (SVD) to decompose the weight matrix into a principal component and a residual component, optimizing the principal part using two trainable low-rank matrices, similar to LoRA. LoRA is further developed into more efficient variants such as QLoRA (Dettmers et al., 2023), which reduces memory requirements and improves computational efficiency by quantization. AdaLoRA enhances LoRA by adaptively adjusting the rank of low-rank updates (Zhang et al., 2023).

*ii) Random projection.* Methods in this category employ random projections to further reduce the memory consumption on parameterized projections. Flora (Hao et al., 2024) develops LoRA by substituting one of the two low-rank matrices with a randomly generated matrix, reducing the memory consumption with comparable performance. VeRA (Kopiczko et al., 2024) employs a shared pair of random projections across all layers, fine-tuning the model by training layer-specific scaling vectors.

*iii) Analytic projection.* Both parameterized and random projections may lead to low-quality approximations of gradient or weight updates, as they lack analytic guarantees. In contrast, GaLore (Zhao et al., 2024) introduces analytic projections derived from SVD to ensure that the key components of the gradients are preserved after low-rank projections, thereby offering a more accurate and reliable approximation.

Existing methods for estimating low-rank projections involve a trade-off between approximation error and resource consumption (*e.g.*, on memory and time). Parameterized and random projections lack the accuracy of analytic decomposition, leading to uncertain quality, while analytic methods like SVD, though more precise, are computationally expensive. For instance, in GaLore, SVD accounts for over 80% of the total time consumed during the fine-tuning process. Therefore, a faster and more accurate estimation of low-rank projections is essential to further boost the low-rank adaptation methods.

In this paper, we propose a low-rank adaptation method, GaLore+, which employs cross-head low-rank projection to realize fast and high-quality estimation. Algorithm 1 presents the pseudo-code of integrating GaLore+ into AdamW, with the highlighted improvement over GaLore (Zhao et al., 2024). GaLore+ utilizes a cross-head low-rank projection inspired by the architecture of multi-head attention, where the projection matrices for the gradient of query or key transforms are shared across multiple attention heads. Such sharing reduces the computational complexity of low-rank projection matrices in $h$-head attention from $O(h^3)$ to $O(h)$. Additionally, randomized subspace iteration for SVD is utilized to further reduce computational complexity. Besides, GaLore+ incorporates sparsely coded residuals, enabling a sparse representation of low-rank approximation errors in weight updates, which helps to mitigate estimation inaccuracies caused by the cross-head low-rank projection. We evaluate the proposed GaLore+ on natural language processing and arithmetic reasoning tasks, comparing it against state-of-the-art low-rank adaptation methods.

The main contributions of this work are as follows:

- We introduce cross-head low-rank projection, which reduces computational complexity by sharing projection matrices across multiple query or key projections. Besides, we employ randomized subspace iteration to accelerate the estimation of the projections.
- We mitigate the impact of low-rank approximation errors on weight updates by utilizing sparsely coded residuals for the optimizer's moments, thereby enhancing the quality of the weight updates.
- Experimental results demonstrate that the proposed method surpasses state-of-the-art approaches, including LoRA and GaLore, in fine-tuning LLMs for tasks such as arithmetic reasoning and natural language generation.

## 2 RELATED WORK

**Parameter Efficient Fine-Tuning.** A variety of parameter-efficient fine-tuning methods have emerged in recent years, enabling an increasing number of institutions and researchers to fine-tune LLMs to meet their specific requirements. Adapters (Rebuffi et al., 2017; Houlsby et al., 2019; Lin et al., 2020; Karimi Mahabadi et al., 2021b;a) enable parameter-efficient fine-tuning by inserting trainable layers into LLMs while keeping other layers frozen. However, this approach also introduces additional inference latency. BitFit (Zaken et al., 2021) selectively tunes only the biases within the network, significantly reducing the number of parameters involved in fine-tuning. Prompt tuning achieves parameter-efficient fine-tuning by optimizing a set of new input tokens or prompts for each task (Li & Liang, 2021; Lester et al., 2021; Hambardzumyan et al., 2021; Liu et al., 2023). Hu et al. (2022) introduced LoRA, proposing that weight updates are low-rank and can be expressed as the product of two low-rank matrices. Furthermore, the trainable parameters can be merged with the original weights, eliminating additional inference latency. Recent studies combined parameter-efficient fine-tuning methods with quantization to enhance memory efficiency during the fine-tuning of LLMs (Kwon et al., 2022; Dettmers et al., 2023; Chai et al., 2023; Xu et al., 2023). And DoRA (Liu et al., 2024), or Weight-Decomposed Low-Rank Adaptation, is a parameter-efficient fine-tuning method designed to enhance learning capacity and stability by decomposing pre-trained weights into magnitude and direction components, leveraging LoRA for directional updates, and achieving superior performance across tasks without additional inference costs.

**Parameter Sharing.** Adam-mini partitions the model parameters into blocks based on the structure of the Hessian matrix, assigning a unified second-order moment to all parameters within each block (Zhang et al., 2024). This approach significantly reduces the memory footprint of the second-order moment, thereby decreasing the optimizer's memory usage. From a temporal perspective, GaLore shares the same projection across a fixed number of steps, reducing computational overhead. The above discussion demonstrates that many parameters can be shared during fine-tuning, reducing memory usage or computational complexity.

**Low-Rank plus Sparse Matrix.** Robust Principal Component Analysis (RPCA) decomposes a data matrix into the sum of the product of low-rank matrices and a sparse matrix and has been extensively studied in both theory and applications (Lin et al., 2010; Zhou & Tao, 2011; Liu et al., 2013; Aravkin et al., 2014; Hintermüller & Wu, 2015; Yi et al., 2016; Zhang & Yang, 2018). The recent Robust Adaptation (RoSA) method extends Low-Rank Adaptation (LoRA) by further decomposing weight updates into the product of two low-rank matrices, with an additional sparse matrix (Nikdan et al., 2024). Using an optimizer to update both the low-rank and sparse matrices, RoSA achieves superior performance compared to LoRA.

## 3 PRELIMINARIES

**GaLore.** Conventional PEFT methods, such as LoRA (Hu et al., 2022), reduce the number of parameters for fine-tuning LLMs. However, the fixed low-rank nature of these methods limits the effectiveness of weight updates, resulting in performance inferior to full fine-tuning. GaLore (Zhao et al., 2024) addresses this limitation by leveraging the low-rank characteristics of gradients and projecting them onto low-rank subspace, significantly reducing the memory requirements for fine-tuning LLMs, while still maintaining the capability for full-parameter tuning. This approach enables

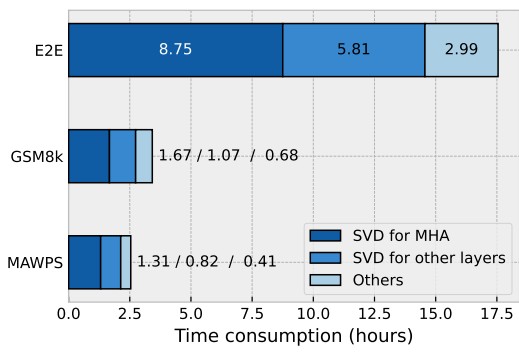
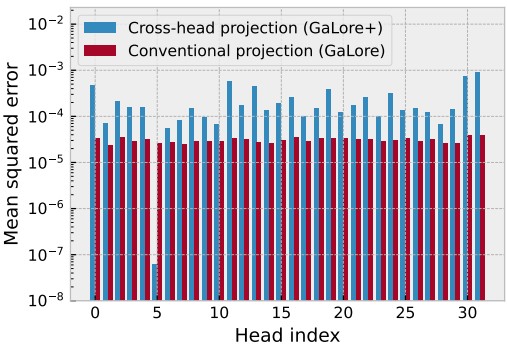

(a) Time consumption of GaLore

(b) Approximation error of low-rank projections

Figure 2: Motivations for cross-head low-rank projection. (a) illustrates the time consumption of SVD and other operations when fine-tuning an LLaMA2-7B model on different datasets with Ga-Lore. MHA is short for multi-head attention. (b) presents the approximation errors of low-rank projection with cross-head projection (*i.e.*, GaLore+) and conventional projection (*i.e.*, GaLore).

pre-training an LLM with 7 billion parameters on a consumer-grade GPU, *i.e.*, NVIDIA RTX 4090 with 24 GB memory. The low-rank projections in GaLore are calculated via SVD and updated at fixed intervals. Thus, the search space for parameters can dynamically change within the full-rank space.

**Transformer.** Multi-head attention is a key component in Transformer (Vaswani et al., 2017), enabling the model to focus on different parts of the input sequence simultaneously. This mechanism employs multiple attention heads, each processing the input sequence independently. Let $\boldsymbol{Q}, \boldsymbol{K}, \boldsymbol{V}$ be $d_{\text{model}}$-dimensional query, key, and value for the multi-head attention, the output of the $i$-th head is

$$\text{head}_i = \text{Attention}(\boldsymbol{Q}\boldsymbol{W}_i^{\boldsymbol{Q}}, \boldsymbol{K}\boldsymbol{W}_i^{\boldsymbol{K}}, \boldsymbol{V}\boldsymbol{W}_i^{\boldsymbol{V}})$$

$$= \text{softmax}\left(\frac{\boldsymbol{Q}\boldsymbol{W}_i^{\boldsymbol{Q}}\left(\boldsymbol{K}\boldsymbol{W}_i^{\boldsymbol{K}}\right)^{\top}}{\sqrt{d_k}}\right)\boldsymbol{V}\boldsymbol{W}_i^{\boldsymbol{V}}, \tag{1}$$

where $\boldsymbol{W}_i^{\boldsymbol{Q}} \in \mathbb{R}^{d_{\text{model}} \times d_k}$, $\boldsymbol{W}_i^{\boldsymbol{K}} \in \mathbb{R}^{d_{\text{model}} \times d_k}$, and $\boldsymbol{W}_i^{\boldsymbol{V}} \in \mathbb{R}^{d_{\text{model}} \times d_v}$ are learned linear transforms. Moreover, the heads are concatenated and further transformed as follows.

$$\text{MultiHead}(\boldsymbol{Q}, \boldsymbol{K}, \boldsymbol{V}) = [\text{head}_1, ..., \text{head}_h]\boldsymbol{W}^{\boldsymbol{O}}, \tag{2}$$

where $\boldsymbol{W}^{\boldsymbol{O}} \in \mathbb{R}^{hd_v \times d_{\text{model}}}$.

## 4 METHODS

To reduce the time consumption of GaLore while improving performance, we propose GaLore+, which introduces two key components, *i.e.*, cross-head low-rank projection and sparsely coded residual. The cross-head low-rank projection enables efficient estimation of projection matrices with reduced computational complexity. Meanwhile, the sparsely coded residual corrects the weight update errors caused by low-rank projection, ensuring more accurate fine-tuning.

### 4.1 CROSS-HEAD LOW-RANK PROJECTION

GaLore demonstrates remarkable performance on PEFT, enabling the training of a 7B LLM on a GPU with just 24GB of memory. However, the SVD employed in GaLore (Zhao et al., 2024) is inherently time-consuming. Figure 2a presents the time consumption for SVD and other operations during the fine-tuning of a LLaMA2-7B model using GaLore. The figure indicates that SVD accounts for more than $80\%$ of the time consumption of the whole fine-tuning process. Moreover, SVD for the multi-head attention (MHA) layers alone takes about half of the fine-tuning process.

### 4.1.1 CROSS-HEAD SIMILARITY FOR SIMPLIFIED SVD

Section 3 has introduced multi-head attention and linear transforms. Note that the three linear transformations across different heads share the same input and can be processed in parallel. Thus, in practical implementations, such as the PyTorch implementation[1], the linear transforms of different heads are concatenated into a single matrix. Here, we denote the *concatenated transforms* as

$$W^Q \triangleq \left[ W_1^Q, W_2^Q, \ldots, W_h^Q \right],$$
$$W^K \triangleq \left[ W_1^K, W_2^K, \ldots, W_h^K \right], \tag{3}$$
$$W^V \triangleq \left[ W_1^V, W_2^V, \ldots, W_h^V \right].$$

Existing low-rank based methods, such as LoRA (Hu et al., 2022) and GaLore (Zhao et al., 2024), conduct low-rank approximation on the updates of the concatenated multi-head transforms rather than on the updates of each individual head. Specifically, GaLore applies SVD to the gradient of a concatenated transform to get the low-rank projection. For instance, the $r$-rank ($r < hd_k$) approximation of the gradient $\nabla W^Q \in \mathbb{R}^{d_{\text{model}} \times hd_k}$ is

$$\nabla W^Q = U\Sigma V \approx U_{:,:r}\Sigma_{:r,:r}V_{:r,:} = PP^\top \nabla W^Q, \tag{4}$$

where $P \triangleq U_{:,:r}$ is the low-rank projection matrix that contains the first $r$ columns of $U$. The low-rank projections for $\nabla W^K$ and $\nabla W^V$ are calculated similarly.

Cordonnier et al. (2020) observed that the query or key transforms of different attention heads within the same layer are similar. The authors show that the concatenated projection is low-rank even though the projections of each head are of high ranks. Therefore, we hypothesize that the gradient $\nabla W_i^Q$ (or $\nabla W_i^K$) of different heads within the same layer are similar. Thus, we can obtain a low-rank projection of the gradient of the concatenated transform $\nabla W^Q$ (or $\nabla W^K$) via SVD on a randomly selected $W_i^Q$ (or $\nabla W_i^K$), *i.e.*,

$$\nabla W_i^Q = U_i\Sigma_i V_i \approx (U_i)_{:,:r}(\Sigma_i)_{:,:r}(V_i)_{:,:r} = P_iP_i^\top \nabla W_i^Q, \tag{5}$$

where $P_i \triangleq (U_i)_{:,:r}$. Thus, the low-rank approximation of $\nabla W^Q$ in the proposed GaLore+ is achieved by with

$$\nabla W^Q \approx P_iP_i^\top \nabla W^Q. \tag{6}$$

Figure 2b illustrates the approximation error of the gradients $\nabla W_i^Q$ with a randomly selected multi-head attention head using the low-rank projections using Equation 4 (vanilla GaLore) and Equation 6 (cross-head low-rank projection). The errors in Equation 6 are larger than those in Equation 4, as shown in Figure 2b. We further introduce sparsely coded residual to reduce for this discrepancy, as discussed in Subsection 4.2. Since the computational complexity of SVD is $O(mn \times \min(m, n))$ for a matrix of $m \times n$ elements, the computational complexity for the SVD operations of $\nabla W^Q \in \mathbb{R}^{d_{\text{model}} \times hd_k}$ and $\nabla W^K \in \mathbb{R}^{d_{\text{model}} \times hd_k}$ can be reduced from $O(h^3d_k^3)$ to $O(hd_k^3)$. As a reference, $h$ is set to 32 in LLaMA2-7B.

### 4.1.2 FAST SVD WITH RANDOMIZED SUBSPACE ITERATION

To further reduce the time consumption on SVD, we adopt the *randomized subspace iteration* algorithm proposed by Halko et al. (2011), which is a fast implementation of SVD. To obtain the $m \times r$ low-rank projection matrix from an $m \times n$ matrix, the randomized subspace iteration can reduce the computational complexity from $O(mn \times \min(m, n))$ to $O(mn \times \log(r))$. Specifically, the computational complexity for the SVD operations of $\nabla W^Q$ and $\nabla W^K$ can be reduced from $O(hd_k^3)$ to $O(hd_k^2 \times \log(r))$.

Furthermore, randomized subspace iteration also reduces memory consumption during fine-tuning. SVD of an $m \times n$ matrix produces three matrices with the shapes of $m \times m$, $\min(m, n)$, and $n \times n$. However, we only use one matrix of $m \times m$ or $n \times n$ to get the low-rank projection matrix. Randomized subspace iteration only produces a $r \times r$ matrix during SVD, which consumes significantly less memory. The experiments in Section 5 demonstrate the advantages of randomized subspace iteration regarding time efficiency and memory usage.

---

[1]https://pytorch.org/docs/stable/generated/torch.nn.MultiheadAttention.html

## 4.2 SPARSELY CODED RESIDUAL

The low-rank projection of the gradients can significantly reduce the memory consumption during fine-tuning, along with the low-rank first- and second-order moments for optimizers. However, the low-rank projection may not always be the main component of practical implementations. For instance, the low-rank projections are updated every hundred steps during fine-tuning due to the high computational complexity of SVD, making it challenging to apply low-rank projections at every step. Moreover, the cross-head low-rank projection introduces increased approximation error due to cross-head sharing and randomized subspace iterations, leading to less accurate SVD results. Consequently, the low-rank first- and second-order moments for the optimizers are also imprecise. To address this, we estimate the residuals using a sparse representation, which improves the quality of the first- and second-order moments.

### 4.2.1 LOW-RANK APPROXIMATION RESIDUAL OF MOMENTS

Let $\boldsymbol{G}_t \in \mathbb{R}^{m \times n}$ be the gradient of $t$-th step, and $\boldsymbol{P}_t \in \mathbb{R}^{m \times r}$ be the low-rank projection of $t$-th step, the residual of low-rank approximation of $\boldsymbol{G}_t$ is

$$\Delta \boldsymbol{G}_t = \boldsymbol{G}_t - \boldsymbol{P}_t \boldsymbol{P}_t^\top \boldsymbol{G}_t. \tag{7}$$

The first- and second-order moments (denoted as $\boldsymbol{M}_t$ and $\boldsymbol{V}_t$, with $\boldsymbol{M}_t, \boldsymbol{V}_t \in \mathbb{R}^{m \times n}$) for common optimizers, such as Adam, AdaGrad, and AdamW, are estimated as

$$\begin{aligned}
\boldsymbol{M}_t &= \beta_1 \boldsymbol{M}_{t-1} + (1 - \beta_1) \boldsymbol{G}_t, \\
\boldsymbol{V}_t &= \beta_2 \boldsymbol{V}_{t-1} + (1 - \beta_2) \boldsymbol{G}_t \odot \boldsymbol{G}_t,
\end{aligned} \tag{8}$$

where $\beta_1$ and $\beta_2$ are decay rates of the moments, and $\odot$ means element-wise multiplication. The low-rank first- and second-order moments (denoted as $\boldsymbol{M}_t'$ and $\boldsymbol{V}_t'$, with $\boldsymbol{M}_t', \boldsymbol{V}_t' \in \mathbb{R}^{r \times n}$) used in GaLore are realized by

$$\begin{aligned}
\boldsymbol{M}_t' &= \beta_1 \boldsymbol{M}_{t-1}' + (1 - \beta_1) \boldsymbol{P}_t^\top \boldsymbol{G}_t, \\
\boldsymbol{V}_t' &= \beta_2 \boldsymbol{V}_{t-1}' + (1 - \beta_2)(\boldsymbol{P}_t^\top \boldsymbol{G}_t) \odot (\boldsymbol{P}_t^\top \boldsymbol{G}_t).
\end{aligned} \tag{9}$$

The low-rank approximation residuals of the moments are

$$\Delta \boldsymbol{M}_t = \boldsymbol{M}_t - \boldsymbol{P}_t \boldsymbol{M}_t', \ \Delta \boldsymbol{V}_t = \boldsymbol{V}_t - \boldsymbol{P}_t \boldsymbol{V}_t'. \tag{10}$$

Equation 10 can be extended into the following form considering Equation 7, 8, and 9,

$$\begin{aligned}
\Delta \boldsymbol{M}_t &\approx \beta_1 \Delta \boldsymbol{M}_{t-1} + (1 - \beta_1) \Delta \boldsymbol{G}_t, \\
\Delta \boldsymbol{V}_t &\approx \beta_2 \Delta \boldsymbol{V}_{t-1} + 2(1 - \beta_2)(\boldsymbol{P}_t \boldsymbol{P}_t^\top \boldsymbol{G}_t) \odot \Delta \boldsymbol{G}_t,
\end{aligned} \tag{11}$$

The detailed inference is provided in Appendix A. Equation 11 depicts the evolution of the low-rank approximation residual of the moments during fine-tuning. Thus, we employ two additional variables in fine-tuning as the estimate of the low-rank approximation residual, and the two variables update following Equation 11.

Furthermore, the approximation residual of the moments leads to bias in parameter updates. The bias formulation depends on the specific form of the optimizer, and we use AdamW as an example. Let $\Delta \boldsymbol{W}_t$ and $\Delta \boldsymbol{W}_t'$ be the full-rank parameter update and the low-rank reconstruction of $\boldsymbol{W}$ at step $t$, then the following equations hold.

$$\Delta \boldsymbol{W}_t = \frac{\boldsymbol{M}_t / (1 - \beta_1^t)}{\sqrt{\boldsymbol{V}_t / (1 - \beta_2^t)} + \epsilon}, \ \Delta \boldsymbol{W}_t' = \frac{\boldsymbol{P}_t \boldsymbol{M}_t' / (1 - \beta_1^t)}{\sqrt{\boldsymbol{P}_t \boldsymbol{V}_t' / (1 - \beta_2^t)} + \epsilon}. \tag{12}$$

Then the low-rank approximation error of the update, denoted as $\delta_t$, is

$$\delta_t = \Delta \boldsymbol{W}_t - \Delta \boldsymbol{W}_t' \approx \frac{\Delta \boldsymbol{M}_t / (1 - \beta_1^t)}{\sqrt{(\Delta \boldsymbol{V}_t + \boldsymbol{P}_t \boldsymbol{V}_t') / (1 - \beta_2^t)} + \epsilon}. \tag{13}$$

Equation 13 demonstrates the evolution of low-rank approximation error of the update during fine-tuning, which is also used in the proposed sparsely coded residual.

---

**Algorithm 2** Sparsely Coded Residual of AdamW

---
1: **given** time step $t$, gradient $\boldsymbol{G}_t \in \mathbb{R}^{m \times n}$, sparse indexing matrix $\boldsymbol{L} \in \mathbb{R}^{m \times n}$, low-rank projection $\boldsymbol{P}_t \in \mathbb{R}^{m \times r}$, second-order low-rank moment $\boldsymbol{V}_t' \in \mathbb{R}^{r \times n}$, first- and second-order moment residuals $\Delta \boldsymbol{M}_{t-1}, \Delta \boldsymbol{V}_{t-1} \in \mathbb{R}^{m \times n}$, constant $\epsilon$
2: $\hat{\boldsymbol{G}}_t \leftarrow \boldsymbol{P}_t \boldsymbol{P}_t^\top \boldsymbol{G}_t$
3: $\Delta \boldsymbol{G}_t \leftarrow (\boldsymbol{G}_t - \hat{\boldsymbol{G}}_t) \odot \boldsymbol{L}$
4: $\Delta \boldsymbol{M}_t \leftarrow \beta_1 \Delta \boldsymbol{M}_{t-1} + (1 - \beta_1)\Delta \boldsymbol{G}_t$
5: $\Delta \boldsymbol{V}_t \leftarrow \beta_2 \Delta \boldsymbol{V}_{t-1} + 2(1 - \beta_2)\hat{\boldsymbol{G}}_t \odot \Delta \boldsymbol{G}_t$
6: $\Delta \hat{\boldsymbol{M}}_t \leftarrow \Delta \boldsymbol{M}_t / (1 - \beta_1^t)$
7: $\Delta \hat{\boldsymbol{V}}_t \leftarrow \Delta \boldsymbol{V}_t / (1 - \beta_2^t)$
8: $\delta_t \leftarrow \Delta \hat{\boldsymbol{M}}_t / \left( \sqrt{\boldsymbol{P}_t \boldsymbol{V}_t' / (1 - \beta_2^t) + \Delta \hat{\boldsymbol{V}}_t} + \epsilon \right)$
9: **return** compact residual $\delta_t$

---

### 4.2.2 SPARSE INDEXING MATRIX FOR THE RESIDUALS

We can leverage the residuals in Equation 11 and 13 to improve the quality of the updates during the optimization process. However, the residuals in Equation 11 and 13 are full-rank matrices, *i.e.*, $\Delta \boldsymbol{M}_t, \Delta \boldsymbol{V}_t, \delta_t \in \mathbb{R}^{m \times n}$, consuming enormous memory during fine-tuning. To incorporate the residuals in memory-efficient PEFT methods, we employ sparse representations with a sparse indexing matrix preserving the most significant elements of the residuals.

We introduce a warm-up stage to determine the sparse indexing matrix essential for efficient fine-tuning. This warm-up stage occurs during the first $k$ steps of the fine-tuning process. At the end of the warm-up stage, the positions for the top-1% absolute values in the reconstructed first-order moment (*i.e.*, $\boldsymbol{P}_t \boldsymbol{M}_t'$) are recorded and used to create the sparse indexing matrix. This allows the approximation residuals for both moments and updates to be stored in sparse matrices, significantly reducing memory requirements. During the warm-up stage, the residuals are set to 0.

Algorithm 2 presents the whole process for obtaining the compact residuals at the $t$-th ($t > k$) step of AdamW. Please note that the proposed method can also be incorporated into other optimizers with moments, such as Adam and AdaGrad, to improve the quality of updates with low-rank approximation.

## 5 EXPERIMENTS

In this section, we present a series of experiments to evaluate the effectiveness of GaLore+. We compare our proposed method against a range of baselines in fine-tuning the LLaMA2-7B and LLaMA2-13B models (Touvron et al., 2023), specifically focusing on tasks related to arithmetic reasoning and natural language generation, to assess its overall performance.

**Implementation Details.** We fine-tune all layers of the LLaMA2-7B and LLaMA2-13B models, adding sparsely coded residuals only in the query and key projections, and load the parameters in `bfloat16` format. For the arithmetic reasoning task, we evaluate the accuracy on the test set, while for the natural language generation task, we measure both the similarity and quality of the generated text compared to the reference text. All these experiments on LLaMA2-7B are carried out on an NVIDIA RTX 4090 GPU with 24GB of memory, using the Llama-Factory framework (Zheng et al., 2024) for implementation. Additionally, experiments on LLaMA2-13B are conducted on an NVIDIA A800 GPU.

**Baselines.** We apply GaLore+ with the following baseline methods:

*i) LoRA* (Hu et al., 2022) is a lightweight adaptation method. It achieves efficient model adaptation by freezing the backbone network and only optimizing low-rank adapters.

*ii) GaLore* (Zhao et al., 2024) is a memory-efficient full-parameter fine-tuning method that significantly reduces memory usage by projecting gradients onto a low-rank subspace.

Table 1: Comparison of fine-tuning LLaMA2-7B with different PEFT methods on the GSM8k and MAWPS datasets. The metrics for fine-tuning time (Time) and memory consumption (Mem.) are 'hours' and 'GB'.

| Datasets | | GSM8k | | | MAWPS | | |
|---|---|---|---|---|---|---|---|
| Methods | Rank | Time↓ | Mem.↓ | Acc. (%)↑ | Time↓ | Mem.↓ | Acc. (%)↑ |
| LoRA | 32 | 0.53 | 15.65 | 23.30 | 0.40 | **14.36** | 45.80 |
| DoRA | 32 | 1.15 | **15.01** | 21.08 | 0.69 | 15.01 | 44.96 |
| GaLore | 32 | 3.48 | 15.42 | 26.46 | 2.59 | 15.15 | 58.40 |
| **GaLore+** | 32 | 0.88 | 15.23 | **29.11** | 0.62 | 14.91 | **60.50** |
| LoRA | 128 | 0.53 | 16.99 | 30.78 | 0.45 | 15.64 | 65.97 |
| DoRA | 128 | 1.18 | 16.17 | 29.36 | 0.72 | 16.17 | 66.81 |
| GaLore | 128 | 3.50 | 16.06 | 33.66 | 2.61 | 15.79 | 64.29 |
| **GaLore+** | 128 | 0.92 | **15.73** | **34.65** | 0.62 | **15.44** | **67.64** |
| LoRA | 256 | 0.55 | 18.79 | 33.36 | 0.4 | 17.60 | 61.76 |
| DoRA | 256 | 1.24 | 18.12 | 33.59 | 0.76 | 18.12 | 62.18 |
| GaLore | 256 | 3.53 | 16.66 | 31.92 | 2.66 | 16.39 | 63.03 |
| **GaLore+** | 256 | 0.92 | **15.74** | **33.81** | 0.69 | **16.12** | **65.55** |

Table 2: Comparison of fine-tuning LLaMA2-13B with different PEFT methods on the GSM8k and MAWPS datasets. The metrics for fine-tuning time (Time) and memory consumption (Mem.) are 'hours' and 'GB'.

| Datasets | | GSM8k | | | MAWPS | | |
|---|---|---|---|---|---|---|---|
| Methods | Rank | Time↓ | Mem.↓ | Acc. (%)↑ | Time↓ | Mem.↓ | Acc. (%)↑ |
| LoRA | 32 | 0.73 | 27.96 | 32.01 | 0.50 | **26.50** | 55.04 |
| DoRA | 32 | 1.68 | **27.04** | 32.12 | 1.03 | 26.57 | 53.78 |
| GaLore | 32 | 7.08 | 28.14 | 36.24 | 5.34 | 27.94 | 61.34 |
| GaLore+ | 32 | 1.22 | 27.93 | **37.98** | 0.80 | 27.59 | **63.87** |
| LoRA | 128 | 0.77 | 30.11 | 33.99 | 0.53 | 28.77 | 54.62 |
| DoRA | 128 | 1.78 | 28.98 | 32.68 | 1.09 | 28.70 | 54.62 |
| GaLore | 128 | 7.20 | 29.28 | 40.03 | 5.39 | 28.99 | 65.13 |
| GaLore+ | 128 | 1.37 | **28.90** | **41.24** | 0.91 | **28.54** | **69.75** |
| LoRA | 256 | 0.80 | 33.08 | 34.72 | 0.55 | 31.62 | 62.18 |
| DoRA | 256 | 1.83 | 31.88 | 33.59 | 1.14 | 31.72 | 62.18 |
| GaLore | 256 | 7.25 | 30.06 | 37.15 | 5.50 | 29.87 | 62.18 |
| GaLore+ | 256 | 1.54 | **29.79** | **42.20** | 1.06 | **29.45** | **66.39** |

*iii) DoRA (Liu et al., 2024)* builds on LoRA by decomposing weights into magnitude and direction, enhancing learning efficiency and stability without increasing inference costs.

## 5.1 ARITHMETIC REASONING

**Setups.** For the arithmetic reasoning task, we utilize the GSM8k and MAWPS datasets to fine-tune and evaluate the models. The GSM8k dataset was created by experienced problem writers and in a variety of linguistic forms (Cobbe et al., 2021). The MAWPS dataset contains arithmetic and algebraic word problems of varying levels of complexity (Koncel-Kedziorski et al., 2016). We use accuracy as the metric to evaluate the effectiveness of different fine-tuning methods. We set the rank $r$ to 32, 128 and 256 for LoRA, DoRA, GaLore, and GaLore+. Detailed hyperparameter settings are provided in the Appendix B.

**Main Results.** Table 1 and 2 compares the performance of GaLore+ with other PEFT methods on the LLaMA2 models. On the GSM8k dataset, GaLore+ outperforms other PEFT methods in accuracy at ranks. For instance, at rank 128, GaLore+ achieves an accuracy of 34.65% on the GSM8k dataset , when fine-tuning LLaMA2-7B, surpassing the second-best result of 33.66%. Similar trends are observed on the MAWPS dataset, further demonstrating the superior performance of GaLore+.

Table 3: Comparison of fine-tuning LLaMA2-7B with different PEFT methods on the E2E dataset. The metrics for fine-tuning time (Time) and memory consumption (Mem.) are 'hours' and 'GB'.

| Methods | Rank | Time↓ | Mem.↓ | BLEU↑ | NIST↑ | MET↑ | ROUGE-1/2/L↑ | CIDEr↑ |
|---------|------|-------|-------|-------|-------|------|--------------|--------|
| LoRA | 32 | 2.94 | 14.17 | 62.47 | 4.58 | 35.07 | 69.16 / 38.00 / 46.24 | 1.46 |
| DoRA | 32 | 5.14 | **14.13** | 62.63 | 4.59 | 35.11 | 69.25 / 38.07 / 46.31 | 1.47 |
| GaLore | 32 | 18.57 | 15.15 | 64.95 | 4.96 | 36.92 | 70.99 / 40.77 / 49.15 | 1.77 |
| GaLore+ | 32 | 4.41 | 14.92 | **65.15** | **4.97** | **37.32** | **71.17 / 41.06 / 49.38** | **1.78** |
| LoRA | 128 | 3.02 | **15.43** | 64.60 | 4.86 | 36.74 | 70.67 / 40.20 / 48.54 | 1.70 |
| DoRA | 128 | 5.27 | **15.43** | **64.77** | 4.86 | **36.77** | 70.72 / 40.20/ 48.62 | 1.72 |
| GaLore | 128 | 18.45 | 15.79 | 64.22 | 5.14 | 36.50 | 70.99 / 41.19 / 49.37 | 1.80 |
| GaLore+ | 128 | 4.52 | **15.43** | 64.22 | **5.16** | 36.66 | **71.07 / 41.54 / 49.71** | **1.82** |
| LoRA | 256 | 3.01 | 17.43 | 64.93 | 4.94 | 36.81 | 70.92 / 40.66 / 49.01 | 1.76 |
| DoRA | 256 | 5.61 | 17.67 | 64.94 | 4.95 | 36.81 | 70.94 / 40.72 / 49.04 | 1.77 |
| GaLore | 256 | 18.42 | 16.39 | 64.97 | 5.10 | 36.84 | **71.19** / 41.34 / 49.26 | 1.82 |
| GaLore+ | 256 | 4.86 | **16.19** | **64.99** | **5.11** | **36.88** | 71.15 / **41.40 / 49.45** | **1.83** |

Table 4: Comparison of fine-tuning LLaMA2-13B with different PEFT methods on the E2E dataset. The metrics for fine-tuning time (Time) and memory consumption (Mem.) are 'hours' and 'GB'.

| Methods | Rank | Time↓ | Mem.↓ | BLEU↑ | NIST↑ | MET↑ | ROUGE-1/2/L↑ | CIDEr↑ |
|---------|------|-------|-------|-------|-------|------|--------------|--------|
| LoRA | 32 | 3.63 | **26.31** | 62.00 | 4.69 | 34.52 | 68.98 / 38.31 / 46.62 | 1.52 |
| DoRA | 32 | 7.55 | 26.44 | 61.99 | 4.70 | 34.58 | 69.00 / 38.40 / 46.65 | 1.51 |
| GaLore | 32 | 38.40 | 27.94 | 65.06 | 4.90 | 37.00 | 70.90 / 40.68 / 48.97 | 1.76 |
| GaLore+ | 32 | 5.90 | 27.58 | **65.48** | **4.92** | **37.31** | **71.12 / 40.89 / 49.13** | **1.79** |
| LoRA | 128 | 3.65 | 28.54 | 64.99 | 4.82 | 36.97 | 70.70 / 40.18 / 48.41 | 1.72 |
| DoRA | 128 | 8.08 | 28.69 | 64.87 | 4.81 | **36.99** | 70.70 / 40.18 / 48.36 | 1.70 |
| GaLore | 128 | 38.12 | 28.99 | 64.35 | 5.05 | 36.48 | 70.81 / 41.04 / 49.17 | 1.78 |
| GaLore+ | 128 | 6.76 | **28.51** | **65.01** | **5.15** | 36.34 | **71.13 / 41.30 / 49.34** | **1.81** |
| LoRA | 256 | 3.89 | 35.97 | **65.27** | 4.89 | 37.12 | 70.89 / 40.60 / 48.79 | 1.74 |
| DoRA | 256 | 8.48 | 31.60 | 65.12 | 4.88 | 37.09 | 70.92 / 40.45 / 48.55 | 1.72 |
| GaLore | 256 | 38.30 | 29.87 | 65.13 | 5.00 | **37.15** | 71.19 / 41.17 / 49.10 | 1.80 |
| GaLore+ | 256 | 7.71 | **29.50** | 64.58 | **5.13** | 36.94 | **71.23 / 41.54 / 49.67** | **1.82** |

## 5.2 NATURAL LANGUAGE GENERATION

**Setups.** For the natural language generation task, we fine-tune and evaluate the model using the E2E dataset (Novikova et al., 2017). We set the rank $r$ to 32, 128 and 256 for thorough comparison. We set the number of training epochs to 1 and set the learning rate to $1 \times 10^{-6}$. We compare the performance of LoRA, DoRA, GaLore, and the proposed GaLore+ with a range of metrics, including peak memory consumption, BLEU, NIST, MET, ROUGE-1/2/L, and CIDEr. For detailed experiment settings, please refer to Appendix B.

**Main Results.** Table 3 and 4 presents the experimental results on the E2E dataset, covering various metrics, with 'Mem.' representing peak memory consumption. Notably, GaLore+ exhibits a competitive or superior performance in most metrics compared to LoRA , DoRA and GaLore. For instance, with rank 128, GaLore+ achieves better results in terms of NIST, ROUGE, and CIDEr scores while maintaining the same memory usage as LoRA , when fine-tuning LLaMA2-7B. Similarly, GaLore+ consistently demonstrates superior performance across most evaluation metrics with a rank of 256. To further validate the capabilities of GaLore +, we fine-tuned LLaMA2-7B and LLaMA2-13B on the CNN/DM and XSum datasets, comparing it against several other PEFT methods. Detailed results are provided in Appendix C.

Additionally, Figure 1 compares the time consumption between GaLore and GaLore+ , when fine-tuning LLaMA2-7B. One of the key advantages of GaLore+ is the negligible time to compute the projection matrices. As a result, the overall fine-tuning time is reduced by more than 70%, significantly accelerating the training process without compromising performance.

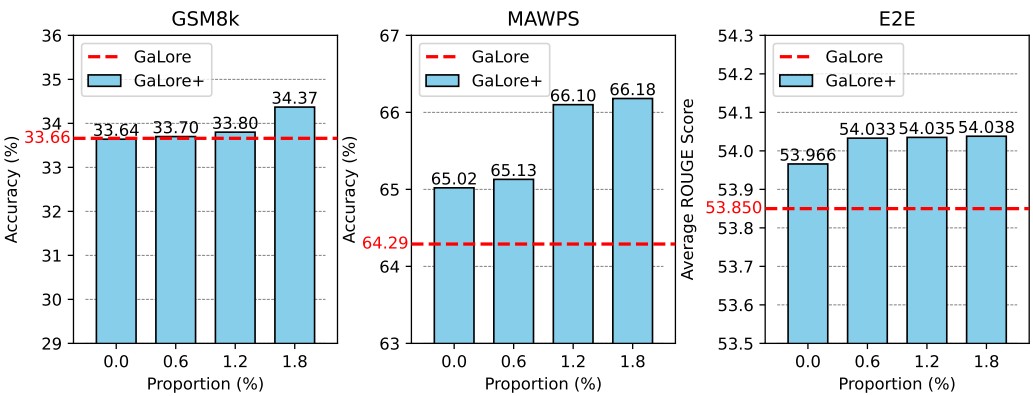

Figure 3: Ablation study on the sparsely coded residual, when fine-tuning LLaMA2-7B. The results are obtained by adjusting the proportion of non-zero elements in the sparse indexing matrix.

## 5.3 ABLATION STUDY

To evaluate the effectiveness of the proposed sparsely coded residual, we conduct experiments on GaLore+ with increasing number of non-zero elements in the sparse indexing matrix. All hyper-parameters except for the sparsity of the residuals are kept unchanged for reasonable ablation. For robustness of the results, experiments were conducted on both LLaMA2-7B and LLaMA3-8B models, with four random seeds set for each experiment. The standard deviations for LLaMA2-7B and the results of LLaMA3-8B are provided in Appendix D.

Figure 3 presents the results of the comparison, suggesting that the residuals with more non-zero elements lead to better performance in most cases which indicates that the sparsely coded residuals indeed mitigate projection-induced errors and enhance the model's performance.

## 6 CONCLUSIONS

In this paper, we propose GaLore+, a low-rank adaptation method with fast yet precise estimation of the low-rank projections. The estimation is achieved by cross-head low-rank projection and random-ized subspace iteration. We further employ a sparsely coded residual to further reduce the error of low-rank approximation, which works on the moments of the optimizer. Experiments on arithmetic reasoning and natural language generation indicate that GaLore+ outperforms existing low-rank adaptation methods, offering superior performance with reduced computational complexity.

However, the proposed method can be further improved in the following aspects.

**Additional theoretical analysis.** Although the experiments validate that sharing low-rank projec-tion matrices across heads in multi-head attention can be efficient, additional theoretical analysis is needed to measure the preciseness of such sharing.

**Improvement on sparsely coded residual.** We employ sparse matrices to store the approximation errors of first- and second-order moments and weight updates. However, all these sparse matrices share the same indexing matrices. Although experiments indicate that the current design is effective, the shared indexing matrices estimated from the first-order moment can be further improved.

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

## A  PROOF OF SPARSELY CODED RESIDUAL

The origin of errors and the calculation of sparsely coded residual are analyzed, using AdamW as an example.

First-order moment (denoted as $\boldsymbol{M}_t$, with $\boldsymbol{M}_t \in \mathbb{R}^{m \times n}$) can be regarded as the sum of first-order moment projected back onto the original space and the error (denoted as $\Delta \boldsymbol{M}_t$, with $\Delta \boldsymbol{M}_t \in \mathbb{R}^{m \times n}$). Similarly, the gradient (denoted as $\boldsymbol{G}_t$, with $\boldsymbol{G}_t \in \mathbb{R}^{m \times n}$) can be viewed as the sum of the gradient projected back onto the original space and the error (denoted as $\Delta \boldsymbol{G}_t$, with $\Delta \boldsymbol{G}_t \in \mathbb{R}^{m \times n}$):

$$\boldsymbol{M}_t = \boldsymbol{P}_t \boldsymbol{P}_t^\top \boldsymbol{M}_t + \Delta \boldsymbol{M}_t, \tag{14}$$

$$\boldsymbol{G}_t = \boldsymbol{P}_t \boldsymbol{P}_t^\top \boldsymbol{G}_t + \Delta \boldsymbol{G}_t, \tag{15}$$

where $\boldsymbol{P}_t$ represents the projection matrix at step $t$. Therefore, the first-order moment updates are shown in Equation 16.

$$\begin{aligned} \boldsymbol{M}_t &= \beta_1 \boldsymbol{M}_{t-1} + (1 - \beta_1) \boldsymbol{G}_t \\ &= \beta_1 \boldsymbol{P}_{t-1} \boldsymbol{P}_{t-1}^\top \boldsymbol{M}_{t-1} + (1 - \beta_1) \boldsymbol{P}_t \boldsymbol{P}_t^\top \boldsymbol{G}_t + \beta_1 \Delta \boldsymbol{M}_{t-1} + (1 - \beta_1) \Delta \boldsymbol{G}_t, \end{aligned} \tag{16}$$

The low-rank first-order moments (denoted as $\boldsymbol{M}_t' = \boldsymbol{P}_t^\top \boldsymbol{M}_t$, with $\boldsymbol{M}_t' \in \mathbb{R}^{r \times n}$) used in GaLore are realized by:

$$\boldsymbol{M}_t' = \beta_1 \boldsymbol{M}_{t-1}' + (1 - \beta_1) \boldsymbol{P}_t^\top \boldsymbol{G}_t. \tag{17}$$

The low-rank approximation error of the first-order moment is

$$\begin{aligned} \Delta \boldsymbol{M}_t &= \boldsymbol{M}_t - \boldsymbol{P}_t \boldsymbol{M}_t' \\ &= \beta_1 (\boldsymbol{P}_{t-1} - \boldsymbol{P}_t) \boldsymbol{P}_{t-1}^\top \boldsymbol{M}_{t-1} + \beta_1 \Delta \boldsymbol{M}_{t-1} + (1 - \beta_1) \Delta \boldsymbol{G}_t. \end{aligned} \tag{18}$$

The projection matrix $\boldsymbol{P}_t$ is updated every few hundred steps, thus, outside of these updates, $\boldsymbol{P}_{t-1}$ remains equal to $\boldsymbol{P}_t$. When $\boldsymbol{P}_t$ is updated, the weight updates $\Delta \boldsymbol{W}_t$ are so small that $\boldsymbol{P}_{t-1} \approx \boldsymbol{P}_t$, similar to the discussion in Section 4.1.1. Therefore, the low-rank approximation error of the first-order moment is

$$\Delta \boldsymbol{M}_t \approx \beta_1 \Delta \boldsymbol{M}_{t-1} + (1 - \beta_1) \Delta \boldsymbol{G}_t. \tag{19}$$

Similarly, the update process for the second-order moment is as follows:

$$\boldsymbol{V}_t = \boldsymbol{P}_t \boldsymbol{P}_t^\top \boldsymbol{V}_t + \Delta \boldsymbol{V}_t, \tag{20}$$

$$\begin{aligned} \boldsymbol{V}_t =& \beta_2 \boldsymbol{V}_{t-1} + (1 - \beta_2) \boldsymbol{G}_t \odot \boldsymbol{G}_t \\ =& \beta_2 \boldsymbol{P}_{t-1} \boldsymbol{P}_{t-1}^\top \boldsymbol{V}_{t-1} + (1 - \beta_2)(\boldsymbol{P}_t \boldsymbol{P}_t^\top \boldsymbol{G}_t) \odot (\boldsymbol{P}_t \boldsymbol{P}_t^\top \boldsymbol{G}_t) + \beta_2 \Delta \boldsymbol{V}_{t-1} \\ & + (1 - \beta_2) \left[ \Delta \boldsymbol{G}_t \odot \Delta \boldsymbol{G}_t + 2(\boldsymbol{P}_t \boldsymbol{P}_t^\top \boldsymbol{G}_t) \odot \Delta \boldsymbol{G}_t \right], \end{aligned} \tag{21}$$

$$\boldsymbol{V}_t' = \beta_2 \boldsymbol{V}_{t-1}' + (1 - \beta_2)(\boldsymbol{P}_t^\top \boldsymbol{G}_t) \odot (\boldsymbol{P}_t^\top \boldsymbol{G}_t), \tag{22}$$

$$\begin{aligned} \Delta \boldsymbol{V}_t =& \boldsymbol{V}_t - \boldsymbol{P}_t \boldsymbol{V}_t' \\ \approx& \beta_2 \Delta \boldsymbol{V}_{t-1} + 2(1 - \beta_2)(\boldsymbol{P}_t \boldsymbol{P}_t^\top \boldsymbol{G}_t) \odot \Delta \boldsymbol{G}_t \\ & + (1 - \beta_2) \left[ \Delta \boldsymbol{G}_t \odot \Delta \boldsymbol{G}_t + (\boldsymbol{P}_t \boldsymbol{P}_t^\top \boldsymbol{G}_t) \odot (\boldsymbol{P}_t \boldsymbol{P}_t^\top \boldsymbol{G}_t) - \boldsymbol{P}_t (\boldsymbol{P}_t^\top \boldsymbol{G}_t) \odot (\boldsymbol{P}_t^\top \boldsymbol{G}_t) \right] \\ \approx& \beta_2 \Delta \boldsymbol{V}_{t-1} + 2(1 - \beta_2)(\boldsymbol{P}_t \boldsymbol{P}_t^\top \boldsymbol{G}_t) \odot \Delta \boldsymbol{G}_t \\ & + (1 - \beta_2) \left[ (\boldsymbol{P}_t \boldsymbol{P}_t^\top \boldsymbol{G}_t) \odot (\boldsymbol{P}_t \boldsymbol{P}_t^\top \boldsymbol{G}_t) - \boldsymbol{P}_t (\boldsymbol{P}_t^\top \boldsymbol{G}_t) \odot (\boldsymbol{P}_t^\top \boldsymbol{G}_t) \right]. \end{aligned} \tag{23}$$

$\Delta \boldsymbol{V}_t$ can be approximated as 0. The expression $(\boldsymbol{P}_t \boldsymbol{P}_t^\top \boldsymbol{G}_t) \odot (\boldsymbol{P}_t \boldsymbol{P}_t^\top \boldsymbol{G}_t) - \boldsymbol{P}_t (\boldsymbol{P}_t^\top \boldsymbol{G}_t) \odot (\boldsymbol{P}_t^\top \boldsymbol{G}_t)$ is only a part of $\Delta \boldsymbol{V}_t$, and its value is quite small. Moreover, considering the significant computational burden associated with this part, we neglect its calculation:

$$\Delta \boldsymbol{V}_t \approx \beta_2 \Delta \boldsymbol{V}_{t-1} + 2(1 - \beta_2)(\boldsymbol{P}_t \boldsymbol{P}_t^\top \boldsymbol{G}_t) \odot \Delta \boldsymbol{G}_t.$$

Thus, the parameter update can be expressed as follows:

$$
\begin{aligned}
\Delta \boldsymbol{W}_t &= \frac{\boldsymbol{M}_t/(1-\beta_1^t)}{\sqrt{\boldsymbol{V}_t/(1-\beta_2^t)}+\epsilon} \\
&= \frac{\Delta \hat{\boldsymbol{M}}_t + \hat{\boldsymbol{M}}_t}{\sqrt{\Delta \hat{\boldsymbol{V}}_t + \hat{\boldsymbol{V}}_t}+\epsilon} \\
&= \frac{\hat{\boldsymbol{M}}_t}{\sqrt{\hat{\boldsymbol{V}}_t}+\epsilon} + \frac{\hat{\boldsymbol{M}}_t\left(\sqrt{\hat{\boldsymbol{V}}_t}-\sqrt{\Delta \hat{\boldsymbol{V}}_t + \hat{\boldsymbol{V}}_t}\right)+\Delta \hat{\boldsymbol{M}}_t\left(\sqrt{\hat{\boldsymbol{V}}_t}+\epsilon\right)}{\left(\sqrt{\Delta \hat{\boldsymbol{V}}_t + \hat{\boldsymbol{V}}_t}+\epsilon\right)\left(\sqrt{\hat{\boldsymbol{V}}_t}+\epsilon\right)},
\end{aligned}
\tag{24}
$$

where

$$
\Delta \hat{\boldsymbol{M}}_t = \Delta \boldsymbol{M}_t/\left(1-\beta_1^t\right), \; \hat{\boldsymbol{M}}_t = \boldsymbol{P}_t \boldsymbol{M}_t'/\left(1-\beta_1^t\right),
$$
$$
\Delta \hat{\boldsymbol{V}}_t = \Delta \boldsymbol{V}_t/\left(1-\beta_2^t\right), \; \hat{\boldsymbol{V}}_t = \boldsymbol{P}_t \boldsymbol{V}_t'/\left(1-\beta_2^t\right),
$$

$\Delta \hat{\boldsymbol{V}}_t$ can be approximated as 0, so the Equation 24 can be approximated as:

$$
\Delta \boldsymbol{W}_t \approx \frac{\hat{\boldsymbol{M}}_t}{\sqrt{\hat{\boldsymbol{V}}_t}+\epsilon} + \frac{\Delta \hat{\boldsymbol{M}}_t}{\sqrt{\Delta \hat{\boldsymbol{V}}_t + \hat{\boldsymbol{V}}_t}+\epsilon} = \frac{\hat{\boldsymbol{M}}_t}{\sqrt{\hat{\boldsymbol{V}}_t}+\epsilon} + \delta_t,
\tag{25}
$$

where $\delta_t$ is the residual neglected by GaLore:

$$
\delta_t = \frac{\Delta \hat{\boldsymbol{M}}_t}{\sqrt{\Delta \hat{\boldsymbol{V}}_t + \hat{\boldsymbol{V}}_t}+\epsilon}.
\tag{26}
$$

## B    DETAILS OF EXPERIMENT

We fine-tuned the LLaMA2-7B model to validate the effectiveness of GaLore+. All experiments were conducted on an NVIDIA RTX 4090 GPU using the Llama-Factory framework. Detailed hyperparameter settings are provided in Table 5.

Table 5: Hyperparameters on three benchmarks for the LLaMA2-7B model.

| Dataset | E2E | | | GSM8k | | | MAWPS | | |
|---|---|---|---|---|---|---|---|---|---|
| Rank | 32 | 128 | 256 | 32 | 128 | 256 | 32 | 128 | 256 |
| Batch Size | 1 | | | 1 | | | 1 | | |
| Epochs | 1 | | | 1 | | | 3 | | |
| LearningRate | 1E-06 | | | 1E-06 | | | 1E-06 | | |
| LR Schedul | cosine | | | cosine | | | cosine | | |
| $\alpha$ | 32 | 128 | 64 | 32 | 64 | 32 | 16 | 32 | 32 |
| Optimizer | AdamW8bit | | | AdamW8bit | | | AdamW8bit | | |
| Residual Ratio | 1.2% | | | 1.2% | | | 1.2% | | |

We fine-tuned the LLaMA2-13B model to validate the effectiveness of GaLore+. All experiments were conducted on an NVIDIA A800 GPU using the Llama-Factory framework. Detailed hyperparameter settings are provided in Table 6.

## C    RESULTS OF CNN/DM AND XSUM DATASETS

We have conducted additional experiments with low-rank settings ($r = 32$) to investigate the effectiveness of GaLore + under low-rank conditions. We have included two additional natural language generation (NLG) datasets (i.e., CNN/DM and XSum), two foundation models (i.e., LLaMA2-7B and LLaMA2-13B), and state-of-the-art PEFT methods as the baseline.

As shown in Table 7, GaLore + consistently achieves comparable or superior results across various scenarios. Experimental results show that the proposed method is specifically effective in low-data fine-tuning scenarios, which outperforms LoRA and other fine-tuning methods significantly.

Table 6: Hyperparameters on three benchmarks for the LLaMA2-13B model.

| Dataset | E2E | | | GSM8k | | | MAWPS | | |
|---|---|---|---|---|---|---|---|---|---|
| Rank | 32 | 128 | 256 | 32 | 128 | 256 | 32 | 128 | 256 |
| Batch Size | 1 | | | 1 | | | 1 | | |
| Epochs | 1 | | | 1 | | | 3 | | |
| Learning Rate | 1E-06 | | | 1E-06 | | | 1E-06 | | |
| LR Schedule | cosine | | | cosine | | | cosine | | |
| $\alpha$ | 32 | 64 | 64 | 32 | | | 32 | 64 | 64 |
| Optimizer | AdamW8bit | | | AdamW8bit | | | AdamW8bit | | |
| Residual Ratio | 1.2% | | | 1.2% | | | 1.2% | | |

Table 7: Comparison of fine-tuning LLaMA2-7B and LLaMA2-13B with different PEFT methods at rank 32 on the CNN/DM and XSum datasets. The metrics for fine-tuning time is 'hours'. R-1, R-2, and R-L are abbreviations for ROUGE-1, ROUGE-2, and ROUGE-L, respectively.

| Models | Methods | CNN/DM | | | | XSum | | | |
|---|---|---|---|---|---|---|---|---|---|
| | | Time↓ | R-1↑ | R-2↑ | R-L↑ | Time↓ | R-1↑ | R-2↑ | R-L↑ |
| LLaMA2-7B | LoRA | 0.37 | 31.68 | 11.45 | 21.85 | 0.25 | 37.25 | 13.46 | 29.50 |
| | DoRA | 0.68 | 31.98 | 11.57 | 22.01 | 0.50 | 37.52 | 13.53 | 29.91 |
| | GaLore | 1.18 | 33.64 | 12.98 | **24.27** | 1.04 | 40.24 | 16.07 | 32.92 |
| | GaLore+ | 0.70 | **33.76** | **13.02** | 23.40 | 0.36 | **40.36** | **16.25** | **33.00** |
| LLaMA2-13B | LoRA | 0.51 | 32.77 | 12.14 | 22.98 | 0.35 | 39.15 | 15.11 | 31.53 |
| | DoRA | 0.94 | 32.69 | 12.00 | 23.10 | 0.71 | 39.38 | 15.40 | 31.72 |
| | GaLore | 2.23 | 33.65 | 13.06 | 24.33 | 2.03 | 42.54 | **18.08** | 35.15 |
| | GaLore+ | 0.65 | **34.05** | **13.25** | **24.44** | 0.49 | **42.70** | 18.06 | **35.26** |

# D    RESULTS OF ABLATION STUDY

To validate the effectiveness of the sparsely coded residual, we also fine-tuned LLaMA3-8B with residuals incorporated at varying proportions. The experimental results are presented as shown in Figure 4

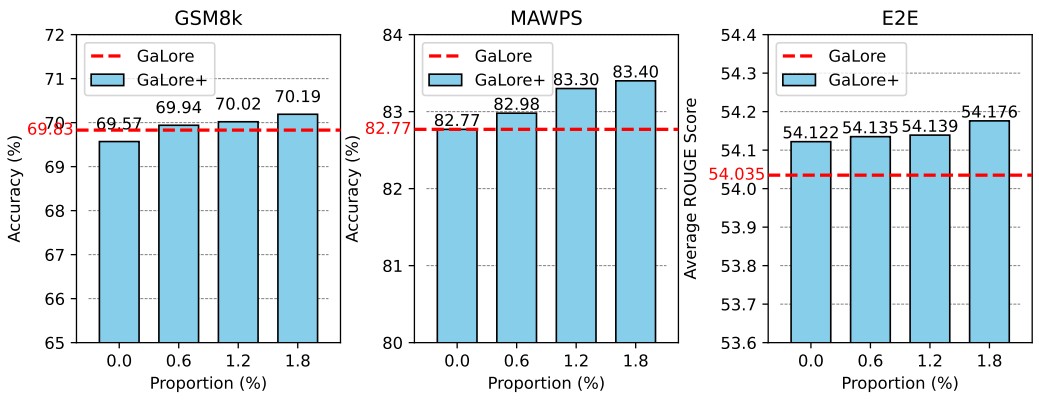

Figure 4: Ablation study on the sparsely coded residual, when fine-tuning LLaMA3-8B. The results are obtained by adjusting the proportion of non-zero elements in the sparse indexing matrix.

The experimental results in Figure 3 and Figure 4 were obtained using 4 random seeds, with the standard deviations reported in Table 8.

Table 8: Ablation study on the sparsely coded residual across different datasets, with proportions 0.0, 0.6, 1.2, and 1.8%.

| Datasets | Models | 0.0 | 0.6 | 1.2 | 1.8 |
|---|---|---|---|---|---|
| GSM8k | LLaMA2-7B | 33.64 (±0.97) | 33.70 (±1.00) | 33.80 (±0.69) | 34.37 (±0.83) |
| | LLaMA3-8B | 69.57 (±0.76) | 69.94 (±0.83) | 70.02 (±0.59) | 70.19 (±0.54) |
| MAWPS | LLaMA2-7B | 65.02 (±2.80) | 65.13 (±0.51) | 66.10 (±1.77) | 66.18 (±1.34) |
| | LLaMA3-8B | 82.77 (±0.59) | 82.98 (±0.63) | 83.30 (±1.00) | 83.40 (±1.91) |
| E2E | LLaMA2-7B | 53.966 (±0.113) | 54.033 (±0.168) | 54.035 (±0.042) | 54.038 (±0.113) |
| | LLaMA3-8B | 54.122 (±0.098) | 54.135 (±0.077) | 54.139 (±0.086) | 54.176 (±0.080) |

