# OpenReview forum: "GaLore$+$: Boosting Low-Rank Adaptation for LLMs with Cross-Head Projection"
_ICLR.cc/2025/Conference — Submitted to ICLR 2025_

### Official Review · Reviewer_R2K7 · 2024-10-28

**Soundness:** 3
**Presentation:** 3
**Contribution:** 3
**Rating:** 6
**Confidence:** 3

**Summary:**

The paper introduces an improved version of GaLore, significantly reducing training time and memory usage. The authors first proposed the cross-head low-rank projection, which obtained the low-ranking gradient projection via the SVD of a randomly selected head rather than that of all heads. They further proposed a mechanism to rectify the first, second moments and the weight updates to reduce the error induced by the low-rank approximation.

**Strengths:**

- GaLore+ showcases its superiority in the arithmetic reasoning task.
- The authors successfully propose a training-accelerating version of Galore, which will reduce both the training resources and improve performance.

**Weaknesses:**

- The empirical results on the natural language generation tasks show insignificant improvements of GaLore+ over LoRa.
- In Line 50, the authors mentioned that the errors in Equation 6 are lower than those in Equation 4, as depicted in Figure 2b. This is because Equation 6 treats all attention heads collectively, thereby not accurately capturing the structure of each individual attention head. Does a lower error indicate a better scenario? Is the error calculated for each head individually or across all heads?
- How does GaLore+ determine the head used for computing low-ranking gradient projection? Based on my understanding, it is selected randomly. Is there any theoretical or empirical rationale behind this choice? If not, at the very least, conducting an ablation study on various head selection strategies is essential.

**Minor errors**:

- Tebble in Line 401.

**Questions:**

NA

---

> ### Author Response · Authors · 2024-11-23
> **Response to R2K7 (part 1)**
>
> Thank you for your valuable feedbacks. Please find our answers and our newly conducted experimental results below.
>
> > The empirical results on the natural language generation tasks show insignificant improvements of GaLore $+$ over LoRa.
>
> **Response:** To further demonstrate the advantages of GaLore $+$, we conducted additional experiments involving a broader range of models and lower rank settings.
>
> *Table R4-1. Fine-tuning LLaMA2-7B and LLaMA2-13B on E2E with diverse ranks.*
>
> | Models     | Methods | Rank | Time(h) $\downarrow$ | Memory (GB) $\downarrow$ | BLEU $\uparrow$ | NIST $\uparrow$ | MET $\uparrow$ | ROUGE-1/2/L $\uparrow$ | CIDEr $\uparrow$ |
> | ---------- | ------- | ---- | ---------------- | ---------------------- | --------------- | --------------- | -------------- | ---------------------- | ---------------- |
> | LLaMA2-7B  | LoRA    | 32   | 2.94  | **14.17** | 62.47 | 4.58 | 35.07 | 69.16/38.00/46.24 | 1.46  |
> |            | GaLore $+$ | 32   | 4.41  | 14.92 | **65.15** | **4.97** | **37.32** | **71.17**/**41.06**/**49.38** | **1.78**  |
> |            | LoRA    | 128  | 3.02  | **15.43** | **64.60** | 4.86 | **36.74** | 70.67/40.20/48.54 | 1.70  |
> |            | GaLore $+$ | 128  | 4.52  | **15.43** | 64.22 | **5.16** | 36.66 | **71.07**/**41.54**/**49.71** | **1.82**  |
> |            | LoRA    | 256  | 3.01  | 17.43 | 64.93 | 4.94 | 36.81 | 70.92/40.66/49.01 | 1.76  |
> |            | GaLore $+$ | 256  | 4.86  | **16.19** | **64.99** | **5.11** | **36.88** | **71.15**/**41.40**/**49.45** | **1.83**  |
> | LLaMA2-13B | LoRA    | 32   | 3.63  | **26.31** | 62.00 | 4.69 | 34.52 | 68.98/38.31/46.62 | 1.52  |
> |            | GaLore $+$ | 32   | 5.90  | 27.58 | **65.48** | **4.92** | **37.31** | **71.12**/**40.89**/**49.13** | **1.79**  |
> |            | LoRA    | 128  | 3.65  | 28.54 | 64.99 | 4.82 | **36.97** | 70.70/40.18/48.41 | 1.72  |
> |            | GaLore $+$ | 128  | 6.76  | **28.51** | **65.01** | **5.15** | 36.34 | **71.13**/**41.30**/**49.34** | **1.81**  |
> |            | LoRA    | 256  | 3.89  | 35.97 | **65.27** | 4.89 | **37.12** | 70.89/40.60/48.79 | 1.74  |
> |            | GaLore $+$ | 256  | 7.71  | **29.50** | 64.58 | **5.13** | 36.94 | **71.23**/**41.54**/**49.67** | **1.82**  |
>
> *Table R4-2. Fine-tuning LLaMA2-7B and LLaMA2-13B on CNN/DM with rank=32.*
>
> | Models     | Methods | Time(h) $\downarrow$ | ROUGE-1 $\uparrow$ | ROUGE-2 $\uparrow$ | ROUGE-L $\uparrow$ |
> | ---------- | ------- | ----------------- | ------------------ | ------------------ | ------------------ |
> | LLaMA2-7B  | LoRA    | 0.37              | 31.68              | 11.45              | 21.85              |
> |            | GaLore $+$ | 0.70              | **33.76**          | **13.02**          | **23.40**              |
> | LLaMA2-13B | LoRA    | 0.51              | 32.77              | 12.14              | 22.98              |
> |            | GaLore $+$ | 0.65              | **34.05**          | **13.25**          | **24.44**          |
>
> *Table R4-3. Fine-tuning LLaMA2-7B and LLaMA2-13B on XSum with rank=32.*
>
> | Models     | Methods | Time(h) $\downarrow$ | ROUGE-1 $\uparrow$ | ROUGE-2 $\uparrow$ | ROUGE-L $\uparrow$ |
> | ---------- | ------- | ----------------- | ------------------ | ------------------ | ------------------ |
> | LLaMA2-7B  | LoRA    | 0.25              | 37.25              | 13.46              | 29.50              |
> |            | GaLore $+$ | 0.36              | **40.36**          | **16.25**          | **33.00**          |
> | LLaMA2-13B | LoRA    | 0.35              | 39.15              | 15.11              | 31.53              |
> |            | GaLore $+$ | 0.49              | **42.70**          | **18.06**              | **35.26**          |
>
> Experimental results show that GaLore $+$ consistently outperforms other methods under the same rank settings, especially when rank=32. Remarkably, in some cases, GaLore $+$ with rank=32 even surpasses the performance of LoRA with rank=128.

---

> ### Author Response · Authors · 2024-11-23
> **Response to R2K7 (part 2)**
>
> > In Line 50, the authors mentioned that the errors in Equation 6 are lower than those in Equation 4, as depicted in Figure 2b. This is because Equation 6 treats all attention heads collectively, thereby not accurately capturing the structure of each individual attention head. Does a lower error indicate a better scenario? Is the error calculated for each head individually or across all heads?
>
> **Response**  We appreciate your thoughtful suggestions. During the rebuttal period, we found the mistake of Figure 2b. We previously mistakenly calculated separate projection matrices for each attention head and used these matrices to compute the approximation error for each head. Thus, the approximation error is lower than that of exact SVD. We have addressed this mistake and revised the manuscript. With the corrected codes, the approximation error of different heads are listed as follows.
>
> *Table R4-5. Approximation Errors of Attention Heads Using Different Projection Methods.*
>
> | Methods                 | head0     | head1     | head2     | head3     | head4     | head5     | head6     | head7     |
> | ----------------------- | --------- | --------- | --------- | --------- | --------- | --------- | --------- | --------- |
> | Exact SVD (in GaLore)   | 0.0000340 | 0.0000236 | 0.0000348 | 0.0000287 | 0.0000314 | 0.0000258 | 0.0000276 | 0.0000247 |
> | Proposed (in GaLore $+$) | 0.0004635 | 0.0000691 | 0.0002145 | 0.0001545 | 0.0001551 | 0.0000001 | 0.0000555 | 0.0000836 |
>
> Please note that we use the low-rank projection matrix of head 5 in the proposed method. Furthermore, GaLore $+$ reduces this approximation error by leveraging the sparsely coded residual. Additionally, due to the cross-head low-rank projection, GaLore $+$ reduces the computational complexity of GaLore's SVD from $O(h^3d_k^3)$ to $O(hd_k^2\times\log(r))$, where $d_k$ is the dimensionality of each attention head, and $h$ is the number of attention heads.
>
> Moreover, we have checked our codes for the main experimental results and there were no mistakes.
>
> > How does GaLore+ determine the head used for computing low-ranking gradient projection? Based on my understanding, it is selected randomly. Is there any theoretical or empirical rationale behind this choice? If not, at the very least, conducting an ablation study on various head selection strategies is essential.
>
> **Response:** We randomly select the head for low-rank gradient projection. We further included an ablation study on selecting different heads for the low-rank projection matrix.
>
> *Table R4-6. Approximation Errors of Attention Heads Using Projection Matrices from Different Heads.*
>
> | Selected head | Head 0    | Head 1    | Head 2    | Head 3    | Head 4    | Head 5    | Head 6    | Head 7    | Average |
> | ------------- | --------- | --------- | --------- | --------- | --------- | --------- | --------- | --------- | ------- |
> | Head 0        | 0.0000001 | 0.0000647 | 0.0001771 | 0.0001431 | 0.0001320 | 0.0000516 | 0.0000542 | 0.0000700 | 0.0000866 |
> | Head 1        | 0.0003566 | 0.0000000 | 0.0002049 | 0.0001562 | 0.0001367 | 0.0000527 | 0.0000511 | 0.0000674 | 0.0001282 |
> | Head 2        | 0.0005905 | 0.0000666 | 0.0000007 | 0.0001478 | 0.0001147 | 0.0000533 | 0.0000544 | 0.0000678 | 0.0001370 |
> | Head 3        | 0.0005192 | 0.0000710 | 0.0002167 | 0.0000001 | 0.0001371 | 0.0000531 | 0.0000550 | 0.0000887 | 0.0001426 |
> | Head 4        | 0.0004661 | 0.0000722 | 0.0001764 | 0.0001189 | 0.0000002 | 0.0000525 | 0.0000558 | 0.0000781 | 0.0001275 |
> | Head 5        | 0.0004635 | 0.0000691 | 0.0002145 | 0.0001545 | 0.0001551 | 0.0000001 | 0.0000555 | 0.0000836 | 0.0001495 |
> | Head 6        | 0.0005912 | 0.0000651 | 0.0002491 | 0.0001579 | 0.0001850 | 0.0000537 | 0.0000001 | 0.0000859 | 0.0001735 |
> | Head 7        | 0.0006270 | 0.0000714 | 0.0002170 | 0.0001384 | 0.0001469 | 0.0000552 | 0.0000556 | 0.0000005 | 0.0001640 |
>
> Table R4-6 shows that choosing different attention heads does not make a significant difference.
>
> > Tebble in Line 401.
>
> **Response:** Thanks for your kind remainder. We have revised the typo.

---

> > ### Comment · Reviewer_R2K7 · 2024-11-24
> >
> > Thanks for the authors' responses. Your answer addresses my concerns and clarifies several points that I initially didn't catch and was confused about. To reflect my more positive attitude towards the revised version, I raised my score to 6. Thanks for your effort in accommodating my questions.

---

### Official Review · Reviewer_zUdp · 2024-11-04

**Soundness:** 2
**Presentation:** 3
**Contribution:** 1
**Rating:** 3
**Confidence:** 4

**Summary:**

The paper presents GaLore+, a new parameter-efficient finetuning (PEFT) technique that extends previous work on GaLore to enhance its time-efficiency as well as downstream performance. To reduce the cost of GaLore taken to compute SVD factors on weight gradients, GaLore+ proposes cross-head low-rank projection which uses a single attention head to compute its low-rank projection, and apply the same projection across all heads. Instead of deterministic SVD, GaLore+ also uses randomized SVD via subspace iterations. Lastly, inspired by previous work on low-rank plus sparse approximators, GaLore+ adds a sparsely coded residual to the low-rank factors, approximating the true gradients better than with only low-rank factors. Experiments using the Llama2-7B model on arithmetic reasoning and natural language generation datasets show that GaLore+ significantly reduces the cost of finetuning in both time and memory.

**Strengths:**

- [S1] **Clear motivation and interesting problem setup.** Parameter-efficient finetuning is a widely used technique for adapting pretrained LLMs, and the techniques introduced in this paper would be a good addition to the LLM finetuning toolkit. The motivation is also crystal clear with preliminary experimental results presented early in the paper: to reduce the time-consumption and enhance performance of GaLore.

**Weaknesses:**

- [W1] **Weak novelty and contribution.** The techniques newly incorporated into GaLore+, which include cross-head low-rank projection, randomized SVD, and sparse residuals, are all essentially from existing work, which significantly weakens the overall novelty of this work. The techniques are also discussed and demonstrated solely under the scope of applying gradient projections (i.e. vanilla GaLore) and thus the contributions are too shallow, and does not seem sufficient to constitute a conference paper at ICLR.
- [W2] **Lacking experimentation.** Even though the paper is mostly empirical, the experimental setup is narrow, covering two arithmetic reasoning datasets and one natural language generation dataset. Considering that GaLore+ can be used essentially anywhere when finetuning pretrained Transformers, it would be great to add (1) more datasets and pretrained models as well as (2) more PEFT baselines (other than LoRA and GaLore) to fully argue that the gain in time-efficiency as well as downstream performance leads to competitive results.
- [W3] **Weak intuition behind cross-head low-rank projection.** The discussion on the cross-head similarity for simplified SVD (Section 4.1.1) is a bit hard to follow. Specifically, how can it be the case that "the gradient of different heads within the same layer are similar" (Lines 236-237), but treating all attention heads as a whole "fails to capture the structure of each individual attention head accurately" (Lines 250-251)?  Shouldn't the hypothesis essentially allow us to treat all attention heads as a whole?

**Questions:**

- [Q1] For Figure 2b, is GaLore+ (in blue) using each $i$-th head to approximate the gradient of the $i$-th head? How can it be that GaLore+ is achieving lower RMSE on all heads, when the rank-$r$ SVD used by vanilla GaLore is known to provide an optimal rank-$r$ approximation in Frobenius norm?
- [Q2] How many random seeds were used for main results in Tables 1 and 2? What were the standard deviations on different performance metrics?
- [Q3] When using randomized subspace iteration to perform fast SVD, how many subspace iterations does GaLore+ use?
- [Q4] Based on Figure 3, it seems the sparsely coded residual does not have a significant impact on downstream performance. Could we compare LoRA and GaLore vs. GaLore+ without sparsely coded residuals to better assess the performance gains?
- Typo in Line 401: Teble 1 $\rightarrow$ Table 1

---

> ### Author Response · Authors · 2024-11-23
> **Response to zUdp (part 1)**
>
> Thank you for your valuable feedbacks. Based on this, we have further conducted more experiments. The detailed responses to the weaknesses and questions are listed as follows.
>
> ### Response to Weaknesses
>
> >  [W1] **Weak novelty and contribution.** The techniques newly incorporated into GaLore $+$, which include cross-head low-rank projection, randomized SVD, and sparse residuals, are all essentially from existing work, which significantly weakens the overall novelty of this work. The techniques are also discussed and demonstrated solely under the scope of applying gradient projections (i.e. vanilla GaLore) and thus the contributions are too shallow, and does not seem sufficient to constitute a conference paper at ICLR.
>
> **Response:** To the best of our knowledge, no prior work has explored cross-head low-rank projection, which is the main contribution of this paper. This approach offers a low-rank projection with minimal error while demonstrating significantly faster calculation compared to existing methods. Our work substantially accelerates vanilla GaLore, and we believe GaLore $+$ offers greater practical usability than its predecessor. Moreover, we believe the proposed methods can also enhance other low-rank projection techniques. We hope that our approach inspires further exploration and refinement of existing low-rank projection methods by the broader research community.
>
> > [W2] **Lacking experimentation.** Even though the paper is mostly empirical, the experimental setup is narrow, covering two arithmetic reasoning datasets and one natural language generation dataset. Considering that GaLore $+$ can be used essentially anywhere when finetuning pretrained Transformers, it would be great to add (1) more datasets and pretrained models as well as (2) more PEFT baselines (other than LoRA and GaLore) to fully argue that the gain in time-efficiency as well as downstream performance leads to competitive results.
>
> **Response:** To further demonstrate the effectiveness of GaLore $+$, we conducted additional experiments with a broader selection of models, datasets, experimental setups, and baselines. The results are presented as follows.
>
> *Table R3-1. Fine-tuning LLaMA2-7B and LLaMA2-13B on GSM8k with diverse ranks.*
>
> | Models     | Methods | Rank | Time(h)$\downarrow$ | Memory (GB) $\downarrow$ | Accuracy (%) $\uparrow$ |
> | ---------- | ------- | ---- | ---------------- | ------------------------ | ----------------------- |
> | LLaMA2-7B  | LoRA    | 32   | 0.53 | 15.65 | 23.30  |
> |            | DoRA    | 32   | 1.15 | **15.01** | 21.08  |
> |            | GaLore  | 32   | 3.48 | 15.42 | 26.46  |
> |            | GaLore $+$ | 32   | 0.88 | 15.23 | **29.11**  |
> |            | LoRA    | 128  | 0.53 | 16.99 | 30.78  |
> |            | DoRA    | 128  | 1.18 | 16.17 | 29.36  |
> |            | GaLore  | 128  | 3.50 | 16.06 | 33.66  |
> |            | GaLore $+$ | 128  | 0.92 | **15.73** | **34.65**  |
> |            | LoRA    | 256  | 0.55 | 18.79 | 33.36  |
> |            | DoRA    | 256  | 1.24 | 18.12 | 33.59  |
> |            | GaLore  | 256  | 3.53 | 16.66 | 31.92  |
> |            | GaLore $+$ | 256  | 0.92 | **15.74** | **33.81**  |
> | LLaMA2-13B | LoRA    | 32   | 0.73 | 27.96 | 32.01  |
> |            | DoRA    | 32   | 1.68 | **27.04** | 32.12  |
> |            | GaLore  | 32   | 7.08 | 28.14 | 36.24  |
> |            | GaLore $+$ | 32   | 1.22 | 27.93 | **37.98**  |
> |            | LoRA    | 128  | 0.77 | 30.11 | 33.99  |
> |            | DoRA    | 128  | 1.78 | 28.98 | 32.68  |
> |            | GaLore  | 128  | 7.20 | 29.28 | 40.03  |
> |            | GaLore $+$ | 128  | 1.37 | **28.90** | **41.24**  |
> |            | LoRA    | 256  | 0.80 | 33.08 | 34.72  |
> |            | DoRA    | 256  | 1.83 | 31.88 | 33.59  |
> |            | GaLore  | 256  | 7.25 | 30.06 | 37.15  |
> |            | GaLore $+$ | 256  | 1.54 | **29.79** | **42.20**  |

---

> > ### Author Response · Authors · 2024-11-23
> > **Response to zUdp (part 2)**
> >
> > *Table R3-2. Fine-tuning LLaMA2-7B and LLaMA2-13B on MAWPS with diverse ranks.*
> >
> > | Models     | Methods | Rank | Time(h) $\downarrow$ | Memory (GB) $\downarrow$ | Accuracy (%) $\uparrow$ |
> > | ---------- | ------- | ---- | ---------------- | ------------------------ | ----------------------- |
> > | LLaMA2-7B  | LoRA    | 32   | 0.40 | **14.36** | 45.80  |
> > |            | DoRA    | 32   | 0.69 | 15.01 | 44.96  |
> > |            | GaLore  | 32   | 2.59 | 15.15 | 58.40  |
> > |            | GaLore $+$ | 32   | 0.62 | 14.91 | **60.50**  |
> > |            | LoRA    | 128  | 0.45 | 15.64 | 65.97  |
> > |            | DoRA    | 128  | 0.72 | 16.17 | 66.81  |
> > |            | GaLore  | 128  | 2.61 | 15.79 | 64.29  |
> > |            | GaLore $+$ | 128  | 0.62 | **15.44** | **67.64**  |
> > |            | LoRA    | 256  | 0.40 | 17.60 | 61.76  |
> > |            | DoRA    | 256  | 0.76 | 18.12 | 62.18  |
> > |            | GaLore  | 256  | 2.66 | 16.39 | 63.03  |
> > |            | GaLore $+$ | 256  | 0.69 | **16.12** | **65.55**  |
> > | LLaMA2-13B | LoRA    | 32   | 0.50 | **26.50** | 55.04  |
> > |            | DoRA    | 32   | 1.03 | 26.57 | 53.78  |
> > |            | GaLore  | 32   | 5.34 | 27.94 | 61.34  |
> > |            | GaLore $+$ | 32   | 0.80 | 27.59 | **63.87**  |
> > |            | LoRA    | 128  | 0.53 | 28.77 | 54.62  |
> > |            | DoRA    | 128  | 1.09 | 28.70 | 54.62  |
> > |            | GaLore  | 128  | 5.39 | 28.99 | 65.13  |
> > |            | GaLore $+$ | 128  | 0.91 | **28.54** | **69.75**  |
> > |            | LoRA    | 256  | 0.55 | 31.62 | 62.18  |
> > |            | DoRA    | 256  | 1.14 | 31.72 | 62.18  |
> > |            | GaLore  | 256  | 5.50 | 29.87 | 62.18  |
> > |            | GaLore $+$ | 256  | 1.06 | **29.45** | **66.39**  |
> >
> > *Table R3-3. Fine-tuning LLaMA2-7B and LLaMA2-13B on E2E with diverse ranks.*
> >
> > | Models     | Methods | Rank | Time(h) $\downarrow$ | Memory (GB) $\downarrow$ | BLEU $\uparrow$ | NIST $\uparrow$ | MET $\uparrow$ | ROUGE-1/2/L $\uparrow$ | CIDEr $\uparrow$ |
> > | ---------- | ------- | ---- | ---------------- | ---------------------- | --------------- | --------------- | -------------- | ---------------------- | ---------------- |
> > | LLaMA2-7B  | LoRA    | 32   | 2.94  | 14.17 | 62.47 | 4.58 | 35.07 | 69.16/38.00/46.24 | 1.46  |
> > |            | DoRA    | 32   | 5.14  | **14.13** | 62.63 | 4.59 | 35.11 | 69.25/38.07/46.31 | 1.47  |
> > |            | GaLore  | 32   | 18.57 | 15.15 | 64.95 | 4.96 | 36.92 | 70.99/40.77/49.15 | 1.77  |
> > |            | GaLore $+$ | 32   | 4.41  | 14.92 | **65.15** | **4.97** | **37.32** | **71.17**/**41.06**/**49.38** | **1.78**  |
> > |            | LoRA    | 128  | 3.02  | **15.43** | 64.60 | 4.86 | 36.74 | 70.67/40.20/48.54 | 1.70  |
> > |            | DoRA    | 128  | 5.27  | **15.43** | **64.77** | 4.86 | **36.77** | 70.72/40.20/48.62 | 1.72  |
> > |            | GaLore  | 128  | 18.45 | 15.79 | 64.22 | 5.14 | 36.50 | 70.99/41.19/49.37 | 1.80  |
> > |            | GaLore $+$ | 128  | 4.52  | **15.43** | 64.22 | **5.16** | 36.66 | **71.07**/**41.54**/**49.71** | **1.82**  |
> > |            | LoRA    | 256  | 3.01  | 17.43 | 64.93 | 4.94 | 36.81 | 70.92/40.66/49.01 | 1.76  |
> > |            | DoRA    | 256  | 5.61  | 17.67 | 64.94 | 4.95 | 36.81 | 70.94/40.72/49.04 | 1.77  |
> > |            | GaLore  | 256  | 18.42 | 16.39 | 64.97 | 5.10 | 36.84 | **71.19**/41.34/49.26 | 1.82  |
> > |            | GaLore $+$ | 256  | 4.86  | **16.19** | **64.99** | **5.11** | **36.88** | 71.15/**41.40**/**49.45** | **1.83**  |
> > | LLaMA2-13B | LoRA    | 32   | 3.63  | **26.31** | 62.00 | 4.69 | 34.52 | 68.98/38.31/46.62 | 1.52  |
> > |            | DoRA    | 32   | 7.55  | 26.44 | 61.99 | 4.70 | 34.58 | 69.00/38.40/46.65 | 1.51  |
> > |            | GaLore  | 32   | 38.40 | 27.94 | 65.06 | 4.90 | 37.00 | 70.90/40.68/48.97 | 1.76  |
> > |            | GaLore $+$ | 32   | 5.90  | 27.58 | **65.48** | **4.92** | **37.31** | **71.12**/**40.89**/**49.13** | **1.79**  |
> > |            | LoRA    | 128  | 3.65  | 28.54 | 64.99 | 4.82 | 36.97 | 70.70/40.18/48.41 | 1.72  |
> > |            | DoRA    | 128  | 8.08  | 28.69 | 64.87 | 4.81 | **36.99** | 70.70/40.18/48.36 | 1.70  |
> > |            | GaLore  | 128  | 38.12 | 28.99 | 64.35 | 5.05 | 36.48 | 70.81/41.04/49.17 | 1.78  |
> > |            | GaLore $+$ | 128  | 6.76  | **28.51** | **65.01** | **5.15** | 36.34 | **71.13**/**41.30**/**49.34** | **1.81**  |
> > |            | LoRA    | 256  | 3.89  | 35.97 | **65.27** | 4.89 | 37.12 | 70.89/40.60/48.79 | 1.74  |
> > |            | DoRA    | 256  | 8.48  | 31.60 | 65.12 | 4.88 | 37.09 | 70.92/40.45/48.55 | 1.72  |
> > |            | GaLore  | 256  | 38.30 | 29.87 | 65.13 | 5.00 | **37.15** | 71.19/41.17/49.10 | 1.80  |
> > |            | GaLore $+$ | 256  | 7.71  | **29.50** | 64.58 | **5.13** | 36.94 | **71.23**/**41.54**/**49.67** | **1.82**  |

---

> > > ### Author Response · Authors · 2024-11-23
> > > **Response to zUdp (part 3)**
> > >
> > > *Table R3-4. Fine-tuning LLaMA2-7B and LLaMA2-13B on CNN/DM with rank=32.*
> > >
> > > | Models     | Methods | Time(h) $\downarrow$ | ROUGE-1 $\uparrow$ | ROUGE-2 $\uparrow$ | ROUGE-L $\uparrow$ |
> > > | ---------- | ------- | ----------------- | ------------------ | ------------------ | ------------------ |
> > > | LLaMA2-7B  | LoRA    | 0.37              | 31.68              | 11.45              | 21.85              |
> > > |            | DoRA    | 0.68              | 31.98              | 11.57              | 22.01              |
> > > |            | GaLore  | 1.18              | 33.64              | 12.98              | **24.27**          |
> > > |            | GaLore $+$ | 0.70              | **33.76**          | **13.02**          | 23.40              |
> > > | LLaMA2-13B | LoRA    | 0.51              | 32.77              | 12.14              | 22.98              |
> > > |            | DoRA    | 0.94              | 32.69              | 12.00              | 23.10              |
> > > |            | GaLore  | 2.23              | 33.65              | 13.06              | 24.33              |
> > > |            | GaLore $+$ | 0.65              | **34.05**          | **13.25**          | **24.44**          |
> > >
> > > *Table R3-5. Fine-tuning LLaMA2-7B and LLaMA2-13B on XSum with rank=32.*
> > >
> > > | Models     | Methods | Time(h) $\downarrow$ | ROUGE-1 $\uparrow$ | ROUGE-2 $\uparrow$ | ROUGE-L $\uparrow$ |
> > > | ---------- | ------- | ----------------- | ------------------ | ------------------ | ------------------ |
> > > | LLaMA2-7B  | LoRA    | 0.25              | 37.25              | 13.46              | 29.50              |
> > > |            | DoRA    | 0.50              | 37.52              | 13.50               | 29.91              |
> > > |            | GaLore  | 1.04              | 40.24              | 16.07              | 32.92              |
> > > |            | GaLore $+$ | 0.36              | **40.36**          | **16.25**          | **33.00**          |
> > > | LLaMA2-13B | LoRA    | 0.35              | 39.15              | 15.11              | 31.53              |
> > > |            | DoRA    | 0.71              | 39.38              | 15.40              | 31.72              |
> > > |            | GaLore  | 2.03              | 42.54              | **18.08**          | 35.15              |
> > > |            | GaLore $+$ | 0.49              | **42.70**          | 18.06              | **35.26**          |
> > >
> > > As shown in the tables, GaLore $+$ consistently achieves comparable or superior results across various scenarios, with its advantages being particularly notable in Arithmetic Reasoning tasks.
> > >
> > > > [W3] **Weak intuition behind cross-head low-rank projection.** The discussion on the cross-head similarity for simplified SVD (Section 4.1.1) is a bit hard to follow. Specifically, how can it be the case that "the gradient of different heads within the same layer are similar" (Lines 236-237), but treating all attention heads as a whole "fails to capture the structure of each individual attention head accurately" (Lines 250-251)? Shouldn't the hypothesis essentially allow us to treat all attention heads as a whole?
> > >
> > > **Response:** We are sorry to the misleading descriptions in the manuscript. We have revised the descriptions. We leverage the similarity among the gradient of different heads to achieve accelerated SVD decomposition.

---

> > > > ### Author Response · Authors · 2024-11-23
> > > > **Response to zUdp (part 4)**
> > > >
> > > > ### Response to Questions
> > > >
> > > > > [Q1] For Figure 2b, is GaLore $+$ (in blue) using each $i$-th head to approximate the gradient of the $i$-th head? How can it be that GaLore $+$ is achieving lower RMSE on all heads, when the rank-$r$ SVD used by vanilla GaLore is known to provide an optimal rank-$r$ approximation in Frobenius norm?
> > > >
> > > > **Response:** We appreciate your thoughtful suggestions. During the rebuttal period, we found the mistake of Figure 2b. We previously mistakenly calculated separate projection matrices for each attention head and used these matrices to compute the approximation error for each head. Thus, the approximation error is lower than that of exact SVD. We have addressed this mistake and revised the manuscript. With the corrected codes, the approximation error of different heads are listed as follows.
> > > >
> > > > *Table R3-6. Approximation Errors of Attention Heads Using Different Projection Methods.*
> > > >
> > > > | Methods                 | head0     | head1     | head2     | head3     | head4     | head5     | head6     | head7     |
> > > > | ----------------------- | --------- | --------- | --------- | --------- | --------- | --------- | --------- | --------- |
> > > > | Exact SVD (in GaLore)   | 0.0000340 | 0.0000236 | 0.0000348 | 0.0000287 | 0.0000314 | 0.0000258 | 0.0000276 | 0.0000247 |
> > > > | Proposed (in GaLore $+$) | 0.0004635 | 0.0000691 | 0.0002145 | 0.0001545 | 0.0001551 | 0.0000001 | 0.0000555 | 0.0000836 |
> > > >
> > > > Please note that we use the low-rank projection matrix of head 5 in the proposed method. Furthermore, GaLore $+$ reduces this approximation error by leveraging the sparsely coded residual. Additionally, due to the cross-head low-rank projection, GaLore $+$ reduces the computational complexity of GaLore's SVD from $O(h^3d_k^3)$ to $O(hd_k^2\times\log(r))$, where $d_k$ is the dimensionality of each attention head, and $h$ is the number of attention heads.
> > > >
> > > > Moreover, we have checked our codes for the main experimental results and there were no mistakes.
> > > >
> > > > > [Q2] How many random seeds were used for main results in Tables 1 and 2? What were the standard deviations on different performance metrics?
> > > >
> > > > **Response:** Due to computational constraints, we only took single-run experiments in Tables 1 and 2. To further validate the performance of GaLore $+$, we expanded the scope of our experiments to include more models, datasets, baselines, and experimental setups, as shown in the Tables R3-1 and R3-5. The extensive experimental results consistently demonstrate the effectiveness of GaLore $+$.
> > > >
> > > > For the ablation studies, we have conducted experiments with multiple runs using four different random seeds, and the corresponding averages and standard deviations are provided below, the results on the GSM8k and MAWPS datasets are reported in terms of accuracy, while the results on the E2E dataset are expressed as the average ROUGE scores.
> > > >
> > > > *Table R3-7. Fine-tuning LLaMA2-7B and LLaMA3-8B on GSM8k with rank=128.*
> > > >
> > > > | Proportion (%)    | 0       |  0.6%       | 1.2%       | 1.8%       |
> > > > | ----------------- | ------- | ----------- | ---------- | ---------- |
> > > > | LLaMA2-7B         | 33.64% ($\pm$ 0.97%) | 33.70% ($\pm$ 1.00%)     | 33.80% ($\pm$ 0.69%)     | 34.37% ($\pm$ 0.83%)    |
> > > > | LLaMA3-8B         | 69.57% ($\pm$ 0.76%) | 69.94% ($\pm$ 0.83%)     | 70.02% ($\pm$ 0.59%)    | 70.19% ($\pm$ 0.54%)    |
> > > >
> > > > *Table R3-8. Fine-tuning LLaMA2-7B and LLaMA3-8B on MAWPS with rank=128.*
> > > >
> > > > | Proportion (%)    | 0       |  0.6%       | 1.2%       | 1.8%        |
> > > > | ----------------- | ------- | ----------- | ---------- | ----------- |
> > > > | LLaMA2-7B         | 65.02% ($\pm$ 2.80%) | 65.13% ($\pm$ 0.51%)     | 66.10% ($\pm$ 1.77%)    | 66.18% ($\pm$ 1.34%)     |
> > > > | LLaMA3-8B         | 82.77% ($\pm$ 0.59%) | 82.98% ($\pm$ 0.63%)     | 83.30% ($\pm$ 1.00%)    | 83.40% ($\pm$ 1.91%)     |
> > > >
> > > > *Table R3-9. Fine-tuning LLaMA2-7B and LLaMA3-8B on E2E with rank=128.*
> > > >
> > > > | Proportion (%)    | 0       |  0.6%       | 1.2%       | 1.8%        |
> > > > | ----------------- | ------- | ----------- | ---------- | ----------- |
> > > > | LLaMA2-7B         | 53.966 ($\pm$ 0.113) | 54.033 ($\pm$ 0.168)     | 54.035 ($\pm$ 0.042)    | 54.038 ($\pm$ 0.113)     |
> > > > | LLaMA3-8B         | 54.122 ($\pm$ 0.098) | 54.135 ($\pm$ 0.077)     | 54.139 ($\pm$ 0.086)    | 54.176 ($\pm$ 0.080)     |
> > > >
> > > > > [Q3] When using randomized subspace iteration to perform fast SVD, how many subspace iterations does GaLore $+$ use?
> > > >
> > > > **Response:** When using randomized subspace iteration, we performed only two iterations. Experimental results show that two iterations are sufficient to achieve a high-quality projection space, with the error after two iterations remaining within 0.1% of the original matrix. Additional iterations provide slight improvements while significantly increase computational cost.

---

> ### Author Response · Authors · 2024-11-23
> **Response to zUdp (part 5)**
>
> > [Q4] Based on Figure 3, it seems the sparsely coded residual does not have a significant impact on downstream performance. Could we compare LoRA and GaLore vs. GaLore $+$ without sparsely coded residuals to better assess the performance gains?
>
> **Response:** Figure 3 in the paper demonstrates that a higher proportion of residuals generally leads to better model performance. Additionally, we compared the effects of different methods, further validating the efficacy of GaLore $+$, the results on the GSM8k and MAWPS datasets are reported in terms of accuracy, while the results on the E2E dataset are expressed as the average ROUGE scores.
>
> *Table R3-10. Fine-tuning LLaMA2-7B and LLaMA3-8B on GSM8k with rank=128.*
>
> | Methods | LoRA  | GaLore | GaLore $+$ (w/o residuals) | GaLore $+$ (w/ residuals) |
> |---------------------------|-------|--------|--------------|--------------|
> | LLaMA2-7B                 | 30.78% | 33.66%  | 33.64% | 33.80% |
> | LLaMA3-8B                 | 54.62% | 69.83%  | 69.57% | 70.02% |
>
> *Table R3-11. Fine-tuning LLaMA2-7B and LLaMA3-8B on MAWPS with rank=128.*
>
> | Methods | LoRA  | GaLore | GaLore $+$ (w/o residuals) | GaLore $+$ (w/ residuals) |
> |--------------------------|-------|--------|---------|--------|
> | LLaMA2-7B                | 65.97% | 64.29%  |  65.02% | 66.10% |
> | LLaMA3-8B                | 67.23% | 82.77%  | 82.77%  | 83.30% |
>
> *Table R3-12. Fine-tuning LLaMA2-7B and LLaMA3-8B on E2E with rank=128.*
>
> | Methods   | LoRA   | GaLore | GaLore $+$ (w/o residuals) | GaLore $+$ (w/ residuals) |
> | --------- | ------ | ------ | ------------------------- | ------------------------ |
> | LLaMA2-7B | 53.137 | 53.850 | 53.966                    | 54.035                   |
> | LLaMA3-8B | 53.557 | 54.035 | 54.122                    | 54.139                   |
>
> Experimental results indicate that GaLore $+$ without sparsely coded residuals still achieves comparable or superior performance compared to other baselines.
>
> > Typo in Line 401: Teble 1 → Table 1
>
> **Response:** Thanks for your kind remainder. We have revised the typo.

---

> ### Author Response · Authors · 2024-11-25
> **Response to zUdp**
>
> Thank you for your time and constructive comments. We would like to know whether our responses have addressed your concerns. Please feel free to comment if there are any further confusions.

---

> ### Comment · Reviewer_zUdp · 2024-11-25
>
> Thank you authors for the commitment taken into preparing the response. Some of my questions have been addressed but I have decided to maintain my score as I still have significant concerns about the experimental results.
>
> - **Variance in ablation study results (Tables R3-7,8, and 9).** Based on these results, it appears almost all performances are overlapping within one standard deviation. In other words, varying the proportion parameter for sparsely coded residual does not lead to statistically significant performance gains for proportion values from 0% up to 1.8%. I am thus not convinced whether the proposed sparsely coded residual has a significant impact on downstream performance. This variance is also not shared in Figure 3 of the main draft, which can mislead readers on the effect of sparse residuals.
> - **Single-seed results for main experiment (Tables 1 and 2).** Connecting to W3 as well as the first weakness mentioned by Reviewer YKpf, this also makes the performance gap between GaLore and GaLore+ somewhat counter-intuitive, since GaLore+ without sparsely coded residual essentially has the same approximability as vanilla GaLore. While this could have been argued empirically instead, the main quantitative results being obtained only from a single random seed fails to support a convincing argument.
>
> In summary, I believe the paper should be much more rigorous in its experimental setup to fully justify the two contributions mentioned in Lines 113-118. I understand this may be difficult to accomplish due to limited time, and therefore I am open to any further discussions on improving the draft from authors as well as other reviewers.

---

> > ### Author Response · Authors · 2024-11-27
> >
> > Thank you for your valuable suggestions. Our experiments are still ongoing, and we will provide the results in the response in the next few days.

---

> > ### Author Response · Authors · 2024-12-01
> > **Response to zUdp (part 6)**
> >
> > We appreciate your engagement during the discussion period. We have conducted additional experiments to address the concerns on the variance of the experiments.
> >
> > >  **Variance in ablation study results (Tables R3-7,8, and 9).** Based on these results, it appears almost all performances are overlapping within one standard deviation. In other words, varying the proportion parameter for sparsely coded residual does not lead to statistically significant performance gains for proportion values from 0% up to 1.8%. I am thus not convinced whether the proposed sparsely coded residual has a significant impact on downstream performance. This variance is also not shared in Figure 3 of the main draft, which can mislead readers on the effect of sparse residuals.
> >
> > **Response:** The experiments presented in Tables R3-7, R3-8, and R3-9 suggest that the average performance of the proposed method improves as the proportion of sparsely coded residuals increases, although such an improvement may not be statistically significant enough. In addition, we have included the variance in Figure 3 in the revised manuscript. However, we are unable to update the draft on OpenReview at this time, as the deadline for uploading new PDFs has passed. Moreover, the sparsely coded residual is only one of the contributions of the proposed method, while the main contribution of this paper is the cross-head low-rank projection, which remarkably reduces the computational complexity for finding effective low-rank projections.
> >
> >
> > >  **Single-seed results for main experiment (Tables 1 and 2).** Connecting to W3 as well as the first weakness mentioned by Reviewer YKpf, this also makes the performance gap between GaLore and GaLore+ somewhat counter-intuitive, since GaLore+ without sparsely coded residual essentially has the same approximability as vanilla GaLore. While this could have been argued empirically instead, the main quantitative results being obtained only from a single random seed fails to support a convincing argument.
> >
> > **Response:** Due to computational resource constraints, we performed single-seed experiments in Tables 1 and 2 of the submitted manuscript. We have complemented the GaLore $+$ experiments with two additional random seeds, formulating three-seed experiments. The results are presented in Tables R3-13, R3-14, and R3-15, with an additional column indicating whether the experimental results of other methods are within one sigma of those of GaLore $+$.
> >
> > *Table R3-13. Fine-tuning LLaMA2-7B and LLaMA2-13B on GSM8k with diverse ranks.*
> >
> > | Models     | Methods | Rank | Time(h)$\downarrow$ | Memory (GB) $\downarrow$ | Accuracy (%) $\uparrow$ | Within 1 $\sigma$ |
> > | ---------- | ------- | ---- | ---------------- | ------------------------ | ----------------------- | ----------------------- |
> > | LLaMA2-7B  | LoRA    | 32   | 0.53 | 15.65 | 23.30  | False |
> > |            | DoRA    | 32   | 1.15 | **15.01** | 21.08  | False |
> > |            | GaLore  | 32   | 3.48 | 15.42 | 26.46  | False |
> > |            | GaLore $+$ | 32   | 0.92 | 15.23 | **28.46 (± 0.48)**  | - |
> > |            | LoRA    | 128  | 0.53 | 16.99 | 30.78  | False |
> > |            | DoRA    | 128  | 1.18 | 16.17 | 29.36  | False |
> > |            | GaLore  | 128  | 3.50 | 16.06 | 33.66  | False |
> > |            | GaLore $+$ | 128  | 0.92 | **15.73** | **34.93 (± 0.68)**  | - |
> > |            | LoRA    | 256  | 0.55 | 18.79 | 33.36  | False |
> > |            | DoRA    | 256  | 1.24 | 18.12 | 33.59  | True |
> > |            | GaLore  | 256  | 3.53 | 16.66 | 31.92  | False |
> > |            | GaLore $+$ | 256  | 0.99 | **15.74** | **33.78 (± 0.41)**  | - |
> > | LLaMA2-13B | LoRA    | 32   | 0.73 | 27.96 | 32.01  | False |
> > |            | DoRA    | 32   | 1.68 | **27.04** | 32.12  | False |
> > |            | GaLore  | 32   | 7.08 | 28.14 | 36.24  | False |
> > |            | GaLore $+$ | 32   | 1.22 | 27.93 | **38.38 (± 0.47)**  | - |
> > |            | LoRA    | 128  | 0.77 | 30.11 | 33.99  | False |
> > |            | DoRA    | 128  | 1.78 | 28.98 | 32.68  | False |
> > |            | GaLore  | 128  | 7.20 | 29.28 | 40.03  | False |
> > |            | GaLore $+$ | 128  | 1.37 | **28.90** | **41.62 (± 0.31)**  | - |
> > |            | LoRA    | 256  | 0.80 | 33.08 | 34.72  | False |
> > |            | DoRA    | 256  | 1.83 | 31.88 | 33.59  | False |
> > |            | GaLore  | 256  | 7.25 | 30.06 | 37.15  | False |
> > |            | GaLore $+$ | 256  | 1.54 | **29.79** | **42.00 (± 0.62)**  | - |

---

> > > ### Author Response · Authors · 2024-12-01
> > > **Response to zUdp (part 7)**
> > >
> > > *Table R3-14. Fine-tuning LLaMA2-7B and LLaMA2-13B on MAWPS with diverse ranks.*
> > >
> > > | Models     | Methods | Rank | Time(h) $\downarrow$ | Memory (GB) $\downarrow$ | Accuracy (%) $\uparrow$ | Within 1 $\sigma$ |
> > > | ---------- | ------- | ---- | ---------------- | ------------------------ | ----------------------- | ----------------------- |
> > > | LLaMA2-7B  | LoRA    | 32   | 0.40 | **14.36** | 45.80  | False |
> > > |            | DoRA    | 32   | 0.69 | 15.01 | 44.96  | False |
> > > |            | GaLore  | 32   | 2.59 | 15.15 | 58.40  | False |
> > > |            | GaLore $+$ | 32   | 0.64 | 14.91 | **59.38 (± 0.79)**  | - |
> > > |            | LoRA    | 128  | 0.45 | 15.64 | 65.97  | False |
> > > |            | DoRA    | 128  | 0.72 | 16.17 | 66.81  | True |
> > > |            | GaLore  | 128  | 2.61 | 15.79 | 64.29  | False |
> > > |            | GaLore $+$ | 128  | 0.62 | **15.44** | **67.09 (± 0.52)**  | - |
> > > |            | LoRA    | 256  | 0.40 | 17.60 | 61.76  | True |
> > > |            | DoRA    | 256  | 0.76 | 18.12 | 62.18  | True |
> > > |            | GaLore  | 256  | 2.66 | 16.39 | 63.03  | True |
> > > |            | GaLore $+$ | 256  | 0.71 | **16.12** | **63.59 (± 5.00)**  | - |
> > > | LLaMA2-13B | LoRA    | 32   | 0.50 | **26.50** | 55.04  | False |
> > > |            | DoRA    | 32   | 1.03 | 26.57 | 53.78  | False |
> > > |            | GaLore  | 32   | 5.34 | 27.94 | 61.34  | False |
> > > |            | GaLore $+$ | 32   | 0.81 | 27.59 | **66.53 (± 1.95)**  | - |
> > > |            | LoRA    | 128  | 0.53 | 28.77 | 54.62  | False |
> > > |            | DoRA    | 128  | 1.09 | 28.70 | 54.62  | False |
> > > |            | GaLore  | 128  | 5.39 | 28.99 | 65.13  | True |
> > > |            | GaLore $+$ | 128  | 1.12 | **28.54** | **66.39 (± 2.47)**  | - |
> > > |            | LoRA    | 256  | 0.55 | 31.62 | 62.18  | True |
> > > |            | DoRA    | 256  | 1.14 | 31.72 | 62.18  | True |
> > > |            | GaLore  | 256  | 5.50 | 29.87 | 62.18  | True |
> > > |            | GaLore $+$ | 256  | 1.05 | **29.45** | **63.16 (± 2.29)**  | - |

---

> ### Author Response · Authors · 2024-12-01
> **Response to zUdp (part 8)**
>
> *Table R3-15. Fine-tuning LLaMA2-7B and LLaMA2-13B on E2E with diverse ranks.*
>
> | Models     | Methods | Rank | Time(h) $\downarrow$ | Memory (GB) $\downarrow$ | BLEU $\uparrow$ | NIST $\uparrow$ | MET $\uparrow$ | ROUGE-1/2/L $\uparrow$ | CIDEr $\uparrow$ | Indicators within 1 $\sigma$ |
> | ---------- | ------- | ---- | ---------------- | ---------------------- | --------------- | --------------- | -------------- | ---------------------- | ---------------- | ---------------- |
> | LLaMA2-7B  | LoRA    | 32   | 2.94  | 14.17 | 62.47 | 4.58 | 35.07 | 69.16/38.00/46.24 | 1.46  | 0 |
> |            | DoRA    | 32   | 5.14  | **14.13** | 62.63 | 4.59 | 35.11 | 69.25/38.07/46.31 | 1.47  | 0 |
> |            | GaLore  | 32   | 18.57 | 15.15 | 64.95 | **4.96** | 36.92 | 70.99/40.77/**49.15** | **1.77**  | 7 |
> |            | GaLore $+$ | 32   | 4.65  | 14.92 | **65.02 (± 0.19)** | **4.96 (± 0.04)** | **37.20 (± 0.28)** | **71.06 (± 0.13)**/**40.89 (± 0.18)**/**49.15 (± 0.24)** | **1.77 (± 0.01)**  | - |
> |            | LoRA    | 128  | 3.02  | **15.43** | 64.60 | 4.86 | 36.74 | 70.67/40.20/48.54 | 1.70  | 2 |
> |            | DoRA    | 128  | 5.27  | **15.43** | **64.77** | 4.86 | 36.77 | 70.72/40.20/48.62 | 1.72  | 2 |
> |            | GaLore  | 128  | 18.45 | 15.79 | 64.22 | **5.14** | 36.50 | 70.99/41.19/49.37 | 1.80  | 2 |
> |            | GaLore $+$ | 128  | 4.82  | **15.43** | 64.66 (± 0.39) | 5.11 (± 0.05) | **36.87 (± 0.20)** | **71.10 (± 0.04)**/**41.46 (± 0.07)**/**49.59 (± 0.11)** | **1.81 (± 0.01)**  | - |
> |            | LoRA    | 256  | 3.01  | 17.43 | 64.93 | 4.94 | 36.81 | 70.92/40.66/49.01 | 1.76  | 0 |
> |            | DoRA    | 256  | 5.61  | 17.67 | 64.94 | 4.95 | 36.81 | 70.94/40.72/49.04 | 1.77  | 1 |
> |            | GaLore  | 256  | 18.42 | 16.39 | 64.97 | **5.10** | 36.84 | **71.19**/41.34/49.26 | **1.82**  | 5 |
> |            | GaLore $+$ | 256  | 5.24  | **16.19** | **65.09 (± 0.15)** | 5.07 (± 0.04) | **37.05 (± 0.15)** | 71.18 (± 0.03)/**41.39 (± 0.07)**/**49.50 (± 0.16)** | **1.82 (± 0.03)**  | - |
> | LLaMA2-13B | LoRA    | 32   | 3.63  | **26.31** | 62.00 | 4.69 | 34.52 | 68.98/38.31/46.62 | 1.52  | 0 |
> |            | DoRA    | 32   | 7.55  | 26.44 | 61.99 | 4.70 | 34.58 | 69.00/38.40/46.65 | 1.51  | 0 |
> |            | GaLore  | 32   | 38.40 | 27.94 | 65.06 | 4.90 | 37.00 | 70.90/40.68/48.97 | **1.76**  | 2 |
> |            | GaLore $+$ | 32   | 6.20  | 27.58 | **65.48 (± 0.00)** | **4.91 (± 0.02)** | **37.39 (± 0.07)** | **71.15 (± 0.03)**/**40.90 (± 0.08)**/**49.12 (± 0.04)** | 1.75 (± 0.01)  | - |
> |            | LoRA    | 128  | 3.65  | 28.54 | **64.99** | 4.82 | 36.97 | 70.70/40.18/48.41 | 1.72  | 2 |
> |            | DoRA    | 128  | 8.08  | 28.69 | 64.87 | 4.81 | **36.99** | 70.70/40.18/48.36 | 1.70  | 2 |
> |            | GaLore  | 128  | 38.12 | **28.99** | 64.35 | 5.05 | 36.48 | 70.81/41.04/49.17 | 1.78  | 4 |
> |            | GaLore $+$ | 128  | 6.78  | 28.51 | 64.97 (± 0.19) | **5.07 (± 0.08)** | 36.86 (± 0.46) | **71.19 (± 0.10)**/**41.13 (± 0.18)**/**49.48 (± 0.16)** | **1.81 (± 0.03)**  | - |
> |            | LoRA    | 256  | 3.89  | 35.97 | **65.27** | 4.89 | 37.12 | 70.89/40.60/48.79 | 1.74  | 0 |
> |            | DoRA    | 256  | 8.48  | 31.60 | 65.12 | 4.88 | 37.09 | 70.92/40.45/48.55 | 1.72  | 0 |
> |            | GaLore  | 256  | 38.30 | 29.87 | 65.13 | 5.00 | **37.15** | **71.19**/41.17/49.10 | 1.80  | 2 |
> |            | GaLore $+$ | 256  | 7.72  | **29.50** | 64.76 (± 0.22) | **5.10 (± 0.02)** | 36.96 (± 0.05) | 71.18 (± 0.06)/**41.50 (± 0.12)**/**49.69 (± 0.08)** | **1.82 (± 0.03)**  | - |
>
> The experimental results show that the proposed method has acceptable variances on the performance of most experiments, and shows statistically significant improvement over the baseline methods. Moreover, we emphasize that the proposed GaLore $+$ has shown improvements on various experimental settings, including five datasets, three ranks, and three base models, compared to three baseline methods, as presented in Tables 1-4 of the revised manuscript.

---

### Official Review · Reviewer_YKpf · 2024-11-04

**Soundness:** 3
**Presentation:** 3
**Contribution:** 2
**Rating:** 3
**Confidence:** 4

**Summary:**

GaLore+ is proposed as an enhancement to GaLore, addressing the computational bottleneck of SVD which consumes over 80% of training time in the original method. The improved method combines cross-head low-rank projection and randomized subspace iteration for faster SVD computation, while introducing sparsely coded residuals to minimize approximation errors in optimizer moments. Experimental results show that GaLore+ achieves 4x faster fine-tuning speed compared to vanilla GaLore while maintaining or improving performance on arithmetic reasoning and natural language generation tasks.

**Strengths:**

This paper presents an upgraded version of GaLore that employs randomized SVD for acceleration and uses Residual of Moments to compensate for potential accuracy loss from the randomized approach.

**Weaknesses:**

While the core idea is straightforward and easy to understand, the paper's primary weakness lies in insufficient experiment, making it difficult for reviewers to properly evaluate the effectiveness of these enhancements.

- Counter-intuitive results require explanation. While the method aims to accelerate training through randomized SVD and use the sparsely coded residuals to reduce the approximation errors, it's fundamentally still an SVD computation. Theoretically, the performance upper bound should be the exact SVD computation. Although speed improvements are understandable, the significant performance gains over exact SVD computation (GaLore) are unexpected and need explanation

- The experimental methodology raises several concerns. The authors only examine cases with large rank values, despite evidence showing that LoRA typically performs best with r=8~32 during Supervised Fine-Tuning (SFT). Additionally, the single-epoch training approach may result in underfitting, potentially skewing the results. The effectiveness of the Sparsely Coded Residual method also remains uncertain, as there's insufficient evidence to show whether the improvements come from this specific component or could be achieved simply through the combination of random SVD and GaLore.

- Another  limitation of this work is the insufficient comparison with relevant baselines. Given that the paper's main contribution centers on accelerating PEFT training, it should include comparisons with other efficient methods like DoRA and FFT LoRA that achieve similar goals with reduced parameter counts and faster training speeds.

[1] DoRA: Weight-Decomposed Low-Rank Adaptation

[2] Parameter-Efficient Fine-Tuning with Discrete Fourier Transform

**Questions:**

To strengthen the paper, the authors should expand their experimental evaluation to include various rank settings, extended training epochs, comprehensive ablation studies, and detailed comparisons with modern PEFT methods. This would provide a more complete and convincing demonstration of their method's advantages.

---

> ### Author Response · Authors · 2024-11-23
> **Response to YKpf (part 1)**
>
> Thanks for your valuable comments. The detailed responses to the weaknesses and questions and weaknesses are listed as follows.
>
> ### Response to Questions
>
> > To strengthen the paper, the authors should expand their experimental evaluation to include various rank settings, extended training epochs, comprehensive ablation studies, and detailed comparisons with modern PEFT methods. This would provide a more complete and convincing demonstration of their method's advantages.
>
> **Response:**  We have conducted additional experiments to address the concerns on various rank settings, extended training epochs, comprehensive ablation studies, and detailed comparisons with modern PEFT methods.
>
> - **Various rank settings and comparisons with modern PEFT methods.**
>
>   We have conducted additional experiments with low-rank settings ($r=32$) to investigate the effectiveness of GaLore $+$ under low-rank conditions. We have included two additional natural language generation (NLG) datasets (*i.e.*, CNN/DM and XSum), a foundation model (*i.e.*, LLaMA2-13B), and a state-of-the-art PEFT method (*i.e.*, DoRA [R-1]) as the baseline, in evaluation. The results on GSM8k, MAWPS, E2E, CNN/DM and XSum are listed in the following five tables.
>
> *Table R2-1. Fine-tuning LLaMA2-7B and LLaMA2-13B on GSM8k with diverse ranks.*
>
> | Models     | Methods | Rank | Time(h)$\downarrow$ | Memory (GB) $\downarrow$ | Accuracy (%) $\uparrow$ |
> | ---------- | ------- | ---- | ---------------- | ------------------------ | ----------------------- |
> | LLaMA2-7B  | LoRA    | 32   | 0.53 | 15.65 | 23.30  |
> |            | DoRA    | 32   | 1.15 | **15.01** | 21.08  |
> |            | GaLore  | 32   | 3.48 | 15.42 | 26.46  |
> |            | GaLore $+$ | 32   | 0.88 | 15.23 | **29.11**  |
> |            | LoRA    | 128  | 0.53 | 16.99 | 30.78  |
> |            | DoRA    | 128  | 1.18 | 16.17 | 29.36  |
> |            | GaLore  | 128  | 3.50 | 16.06 | 33.66  |
> |            | GaLore $+$ | 128  | 0.92 | **15.73** | **34.65**  |
> |            | LoRA    | 256  | 0.55 | 18.79 | 33.36  |
> |            | DoRA    | 256  | 1.24 | 18.12 | 33.59  |
> |            | GaLore  | 256  | 3.53 | 16.66 | 31.92  |
> |            | GaLore $+$ | 256  | 0.92 | **15.74** | **33.81**  |
> | LLaMA2-13B | LoRA    | 32   | 0.73 | 27.96 | 32.01  |
> |            | DoRA    | 32   | 1.68 | **27.04** | 32.12  |
> |            | GaLore  | 32   | 7.08 | 28.14 | 36.24  |
> |            | GaLore $+$ | 32   | 1.22 | 27.93 | **37.98**  |
> |            | LoRA    | 128  | 0.77 | 30.11 | 33.99  |
> |            | DoRA    | 128  | 1.78 | 28.98 | 32.68  |
> |            | GaLore  | 128  | 7.20 | 29.28 | 40.03  |
> |            | GaLore $+$ | 128  | 1.37 | **28.90** | **41.24**  |
> |            | LoRA    | 256  | 0.80 | 33.08 | 34.72  |
> |            | DoRA    | 256  | 1.83 | 31.88 | 33.59  |
> |            | GaLore  | 256  | 7.25 | 30.06 | 37.15  |
> |            | GaLore $+$ | 256  | 1.54 | **29.79** | **42.20**  |
>
> *Table R2-2. Fine-tuning LLaMA2-7B and LLaMA2-13B on MAWPS with diverse ranks.*
>
> | Models     | Methods | Rank | Time(h) $\downarrow$ | Memory (GB) $\downarrow$ | Accuracy (%) $\uparrow$ |
> | ---------- | ------- | ---- | ---------------- | ------------------------ | ----------------------- |
> | LLaMA2-7B  | LoRA    | 32   | 0.40 | **14.36** | 45.80  |
> |            | DoRA    | 32   | 0.69 | 15.01 | 44.96  |
> |            | GaLore  | 32   | 2.59 | 15.15 | 58.40  |
> |            | GaLore $+$ | 32   | 0.62 | 14.91 | **60.50**  |
> |            | LoRA    | 128  | 0.45 | 15.64 | 65.97  |
> |            | DoRA    | 128  | 0.72 | 16.17 | 66.81  |
> |            | GaLore  | 128  | 2.61 | 15.79 | 64.29  |
> |            | GaLore $+$ | 128  | 0.62 | **15.44** | **67.64**  |
> |            | LoRA    | 256  | 0.40 | 17.60 | 61.76  |
> |            | DoRA    | 256  | 0.76 | 18.12 | 62.18  |
> |            | GaLore  | 256  | 2.66 | 16.39 | 63.03  |
> |            | GaLore $+$ | 256  | 0.69 | **16.12** | **65.55**  |
> | LLaMA2-13B | LoRA    | 32   | 0.50 | **26.50** | 55.04  |
> |            | DoRA    | 32   | 1.03 | 26.57 | 53.78  |
> |            | GaLore  | 32   | 5.34 | 27.94 | 61.34  |
> |            | GaLore $+$ | 32   | 0.80 | 27.59 | **63.87**  |
> |            | LoRA    | 128  | 0.53 | 28.77 | 54.62  |
> |            | DoRA    | 128  | 1.09 | 28.70 | 54.62  |
> |            | GaLore  | 128  | 5.39 | 28.99 | 65.13  |
> |            | GaLore $+$ | 128  | 0.91 | **28.54** | **69.75**  |
> |            | LoRA    | 256  | 0.55 | 31.62 | 62.18  |
> |            | DoRA    | 256  | 1.14 | 31.72 | 62.18  |
> |            | GaLore  | 256  | 5.50 | 29.87 | 62.18  |
> |            | GaLore $+$ | 256  | 1.06 | **29.45** | **66.39**  |

---

> > ### Author Response · Authors · 2024-11-23
> > **Response to YKpf (part 2)**
> >
> > *Table R2-3. Fine-tuning LLaMA2-7B and LLaMA2-13B on E2E with diverse ranks.*
> >
> > | Models     | Methods | Rank | Time(h) $\downarrow$ | Memory (GB) $\downarrow$ | BLEU $\uparrow$ | NIST $\uparrow$ | MET $\uparrow$ | ROUGE-1/2/L $\uparrow$ | CIDEr $\uparrow$ |
> > | ---------- | ------- | ---- | ---------------- | ---------------------- | --------------- | --------------- | -------------- | ---------------------- | ---------------- |
> > | LLaMA2-7B  | LoRA    | 32   | 2.94  | 14.17 | 62.47 | 4.58 | 35.07 | 69.16/38.00/46.24 | 1.46  |
> > |            | DoRA    | 32   | 5.14  | **14.13** | 62.63 | 4.59 | 35.11 | 69.25/38.07/46.31 | 1.47  |
> > |            | GaLore  | 32   | 18.57 | 15.15 | 64.95 | 4.96 | 36.92 | 70.99/40.77/49.15 | 1.77  |
> > |            | GaLore $+$ | 32   | 4.41  | 14.92 | **65.15** | **4.97** | **37.32** | **71.17**/**41.06**/**49.38** | **1.78**  |
> > |            | LoRA    | 128  | 3.02  | **15.43** | 64.60 | 4.86 | 36.74 | 70.67/40.20/48.54 | 1.70  |
> > |            | DoRA    | 128  | 5.27  | **15.43** | **64.77** | 4.86 | **36.77** | 70.72/40.20/48.62 | 1.72  |
> > |            | GaLore  | 128  | 18.45 | 15.79 | 64.22 | 5.14 | 36.50 | 70.99/41.19/49.37 | 1.80  |
> > |            | GaLore $+$ | 128  | 4.52  | **15.43** | 64.22 | **5.16** | 36.66 | **71.07**/**41.54**/**49.71** | **1.82**  |
> > |            | LoRA    | 256  | 3.01  | 17.43 | 64.93 | 4.94 | 36.81 | 70.92/40.66/49.01 | 1.76  |
> > |            | DoRA    | 256  | 5.61  | 17.67 | 64.94 | 4.95 | 36.81 | 70.94/40.72/49.04 | 1.77  |
> > |            | GaLore  | 256  | 18.42 | 16.39 | 64.97 | 5.10 | 36.84 | **71.19**/41.34/49.26 | 1.82  |
> > |            | GaLore $+$ | 256  | 4.86  | **16.19** | **64.99** | **5.11** | **36.88** | 71.15/**41.40**/**49.45** | **1.83**  |
> > | LLaMA2-13B | LoRA    | 32   | 3.63  | **26.31** | 62.00 | 4.69 | 34.52 | 68.98/38.31/46.62 | 1.52  |
> > |            | DoRA    | 32   | 7.55  | 26.44 | 61.99 | 4.70 | 34.58 | 69.00/38.40/46.65 | 1.51  |
> > |            | GaLore  | 32   | 38.40 | 27.94 | 65.06 | 4.90 | 37.00 | 70.90/40.68/48.97 | 1.76  |
> > |            | GaLore $+$ | 32   | 5.90  | 27.58 | **65.48** | **4.92** | **37.31** | **71.12**/**40.89**/**49.13** | **1.79**  |
> > |            | LoRA    | 128  | 3.65  | 28.54 | 64.99 | 4.82 | 36.97 | 70.70/40.18/48.41 | 1.72  |
> > |            | DoRA    | 128  | 8.08  | 28.69 | 64.87 | 4.81 | **36.99** | 70.70/40.18/48.36 | 1.70  |
> > |            | GaLore  | 128  | 38.12 | 28.99 | 64.35 | 5.05 | 36.48 | 70.81/41.04/49.17 | 1.78  |
> > |            | GaLore $+$ | 128  | 6.76  | **28.51** | **65.01** | **5.15** | 36.34 | **71.13**/**41.30**/**49.34** | **1.81**  |
> > |            | LoRA    | 256  | 3.89  | 35.97 | **65.27** | 4.89 | 37.12 | 70.89/40.60/48.79 | 1.74  |
> > |            | DoRA    | 256  | 8.48  | 31.60 | 65.12 | 4.88 | 37.09 | 70.92/40.45/48.55 | 1.72  |
> > |            | GaLore  | 256  | 38.30 | 29.87 | 65.13 | 5.00 | **37.15** | 71.19/41.17/49.10 | 1.80  |
> > |            | GaLore $+$ | 256  | 7.71  | **29.50** | 64.58 | **5.13** | 36.94 | **71.23**/**41.54**/**49.67** | **1.82**  |
> >
> > *Table R2-4. Fine-tuning LLaMA2-7B and LLaMA2-13B on CNN/DM with rank=32.*
> >
> > | Models     | Methods | Time(h) $\downarrow$ | ROUGE-1 $\uparrow$ | ROUGE-2 $\uparrow$ | ROUGE-L $\uparrow$ |
> > | ---------- | ------- | ----------------- | ------------------ | ------------------ | ------------------ |
> > | LLaMA2-7B  | LoRA    | 0.37              | 31.68              | 11.45              | 21.85              |
> > |            | DoRA    | 0.68              | 31.98              | 11.57              | 22.01              |
> > |            | GaLore  | 1.18              | 33.64              | 12.98              | **24.27**          |
> > |            | GaLore $+$ | 0.70              | **33.76**          | **13.02**          | 23.40              |
> > | LLaMA2-13B | LoRA    | 0.51              | 32.77              | 12.14              | 22.98              |
> > |            | DoRA    | 0.94              | 32.69              | 12.00              | 23.10              |
> > |            | GaLore  | 2.23              | 33.65              | 13.06              | 24.33              |
> > |            | GaLore $+$ | 0.65              | **34.05**          | **13.25**          | **24.44**          |

---

> ### Author Response · Authors · 2024-11-23
> **Response to YKpf (part 3)**
>
> *Table R2-5. Fine-tuning LLaMA2-7B and LLaMA2-13B on XSum with rank=32.*
>
> | Models     | Methods | Time(h) $\downarrow$ | ROUGE-1 $\uparrow$ | ROUGE-2 $\uparrow$ | ROUGE-L $\uparrow$ |
> | ---------- | ------- | ----------------- | ------------------ | ------------------ | ------------------ |
> | LLaMA2-7B  | LoRA    | 0.25              | 37.25              | 13.46              | 29.50              |
> |            | DoRA    | 0.50              | 37.52              | 13.50               | 29.91              |
> |            | GaLore  | 1.04              | 40.24              | 16.07              | 32.92              |
> |            | GaLore $+$ | 0.36              | **40.36**          | **16.25**          | **33.00**          |
> | LLaMA2-13B | LoRA    | 0.35              | 39.15              | 15.11              | 31.53              |
> |            | DoRA    | 0.71              | 39.38              | 15.40              | 31.72              |
> |            | GaLore  | 2.03              | 42.54              | **18.08**          | 35.15              |
> |            | GaLore $+$ | 0.49              | **42.70**          | 18.06              | **35.26**          |
>
> As shown in Tables R2-1 to R2-5, GaLore $+$ consistently achieves comparable or superior results across various scenarios, with its advantages being particularly notable in Arithmetic Reasoning tasks. Moreover, the proposed method achieves faster training speed than DoRA.
>
> - **Extended training epochs.**
>
>   To prevent catastrophic forgetting in fine-tuned models, we followed the common practice of using a relatively low number of epochs. However, for datasets with particularly small sizes, such as MAWPS, we increased the epoch count to 3 to ensure the model does not underfit.
>
>   Moreover, the loss curves during training indicate that the model has converged, supporting the reasonableness of our epoch settings.
>
>   Additionally, we compared the performance of LLaMA2-7B across different epochs, as shown in Table R2-6.
>
> *Table R2-6. Fine-tuning LLaMA2-7B on E2E with ranks and different epochs.*
>
> | rank | Epoch | ROUGE-1 | ROUGE-2 | ROUGE-L  |
> |------|-------|---------|---------|----------|
> | 32   | 1     | 71.17   | 41.06   | 49.38    |
> |      | 2     | 71.38   | 41.45   | 49.62    |
> |      | 3     | 71.12   | 41.33   | 49.34    |
> | 128  | 1     | 71.07   | 41.54   | 49.70    |
> |      | 2     | 71.20   | 41.44   | 49.69    |
> |      | 3     | 71.09   | 41.23   | 49.33    |
>
> The experimental results in Table R2-6 show that the model performs well with just an one-epoch training, and increasing the number of epochs does not necessarily lead to improved performance.
>
> - **Comprehensive ablation studies.**
>
>   To demonstrate the effectiveness of the sparsely coded residual, we conducted ablation study on the proportion of the residual. When the proportion is set 0, the sparsely coded residual is removed and the PEFT method equals to combining cross-head projection (random SVD) with GaLore. We fine-tuned LLaMA2-7B and LLaMA3-8B models on the GSM8k, MAWPS, and E2E datasets for thorough comparison. The results for GSM8k and MAWPS are reported in terms of accuracy, while those for E2E are measured using the average ROUGE score. The experimental results are presented as follows.
>
> *Table R2-7. Fine-tuning LLaMA2-7B and LLaMA3-8B on GSM8k with rank=128.*
>
> | Proportion (%) | 0      | 0.6%   | 1.2%   | 1.8%   |
> | -------------- | ------ | ------ | ------ | ------ |
> | LLaMA2-7B      | 33.64% | 33.70% | 33.80% | 34.37% |
> | LLaMA3-8B      | 69.57% | 69.94% | 70.02% | 70.19% |
>
> *Table R2-8. Fine-tuning LLaMA2-7B and LLaMA3-8B on MAWPS with rank=128.*
>
> | Proportion (%) | 0      | 0.6%   | 1.2%   | 1.8%   |
> | -------------- | ------ | ------ | ------ | ------ |
> | LLaMA2-7B      | 65.02% | 65.13% | 66.10% | 66.18% |
> | LLaMA3-8B      | 82.77% | 82.98% | 83.30% | 83.40% |
>
> *Table R2-9. Fine-tuning LLaMA2-7B and LLaMA3-8B on E2E with rank=128.*
>
> | Proportion(%)     | 0       |  0.6%       | 1.2%       | 1.8%        |
> | ----------------- | ------- | ----------- | ---------- | ----------- |
> | LLaMA2-7B         | 53.966  | 54.033      | 54.035     | 54.038      |
> | LLaMA3-8B         | 54.122  | 54.135      | 54.139      | 54.176     |
>
> The experimental results above indicate that, the sparsely coded residual can improve the performance. Furthermore, the performance of the model improves progressively as the proportion of residuals increases.

---

> ### Author Response · Authors · 2024-11-23
> **Response to YKpf (part 4)**
>
> ### Response to Weaknesses
>
> > Counter-intuitive results require explanation. While the method aims to accelerate training through randomized SVD and use the sparsely coded residuals to reduce the approximation errors, it's fundamentally still an SVD computation. Theoretically, the performance upper bound should be the exact SVD computation. Although speed improvements are understandable, the significant performance gains over exact SVD computation (GaLore) are unexpected and need explanation
>
> **Response:** We appreciate your thoughtful suggestions. During the rebuttal period, we found the mistake of Figure 2b. We previously mistakenly calculated separate projection matrices for each attention head and used these matrices to compute the approximation error for each head. Thus, the approximation error is lower than that of exact SVD. We have addressed this mistake and revised the manuscript. With the corrected codes, the approximation error of different heads are listed as follows.
>
> *Table R2-10. Approximation Errors of Attention Heads Using Different Projection Methods.*
>
> | Methods                  | head0   | head1   | head2   | head3    | head4    | head5    | head6    | head7    |
> |--------------------------|---------|---------|---------|---------|---------|---------|---------|---------|
> | Exact SVD (in GaLore) | 0.0000340 | 0.0000236 | 0.0000348 | 0.0000287 | 0.0000314 | 0.0000258 | 0.0000276 | 0.0000247 |
> | Proposed (in GaLore $+$) | 0.0004635 | 0.0000691 | 0.0002145 | 0.0001545 | 0.0001551 | 0.0000001 | 0.0000555 | 0.0000836 |
>
> Please note that we use the low-rank projection matrix of head 5 in the proposed method. Furthermore, GaLore $+$ reduces this approximation error by leveraging the sparsely coded residual. Additionally, due to the cross-head low-rank projection, GaLore $+$ reduces the computational complexity of GaLore's SVD from $O(h^3d_k^3)$ to $O(hd_k^2\times\log(r))$, where $d_k$ is the dimensionality of each attention head, and $h$ is the number of attention heads.
>
> Moreover, we have checked our codes for the main experimental results and there were no mistakes.
>
> > The experimental methodology raises several concerns. The authors only examine cases with large rank values, despite evidence showing that LoRA typically performs best with r=8~32 during Supervised Fine-Tuning (SFT). Additionally, the single-epoch training approach may result in underfitting, potentially skewing the results. The effectiveness of the Sparsely Coded Residual method also remains uncertain, as there's insufficient evidence to show whether the improvements come from this specific component or could be achieved simply through the combination of random SVD and GaLore.
>
> **Response:** Please refer to our response to questions.
>
> > Another limitation of this work is the insufficient comparison with relevant baselines. Given that the paper's main contribution centers on accelerating PEFT training, it should include comparisons with other efficient methods like DoRA and FFT LoRA that achieve similar goals with reduced parameter counts and faster training speeds.
>
> **Response:** Please refer to our response to questions.
>
>
> **References**:
>
> [R-1] Shih-Yang Liu, Chien-Yi Wang, Hongxu Yin, Pavlo Molchanov, Yu-Chiang Frank Wang, KwangTing Cheng, and Min-Hung Chen. Dora: Weight-decomposed low-rank adaptation. arXiv preprint arXiv:2402.09353, 2024.

---

> ### Author Response · Authors · 2024-11-25
> **Response to YKpf**
>
> Thank you for your time and constructive comments. We would like to know whether our responses have addressed your concerns. Please feel free to comment if there are any further confusions.

---

### Official Review · Reviewer_HiNY · 2024-11-11

**Soundness:** 3
**Presentation:** 3
**Contribution:** 3
**Rating:** 6
**Confidence:** 4

**Summary:**

This paper introduces GaLore+, an improved version of the GaLore method for memory-efficient fine-tuning of large language models. GaLore+ tackles the time bottleneck in GaLore by utilizing cross-head low-rank projections and randomized subspace iteration for faster singular value decomposition. Additionally, it introduces sparsely coded residuals to minimize errors from low-rank approximation. Experiments on arithmetic reasoning (GSM8k, d MAWPS datasets) and natural language generation (E2E) tasks show that GaLore+ achieves significantly faster fine-tuning (around 4x) while maintaining superior performance compared to the original GaLore.

**Strengths:**

1) the analysis (Figure 2) nicely identifies the issue and motivates the paper
2) the proposed method is sound and the results seems convincing.

**Weaknesses:**

1) the evaluation datasets in experiment study are limited. Eval metrics for NLG(E2E) are not suitable for generative task evaluation.
2) The sensitivity of Hparams (rank, low data regime) is not well studied.

**Questions:**

1) Given LoRA's established effectiveness in low-data fine-tuning scenarios, how does the proposed method perform under similar low-rank, data-constrained conditions?
2) Do the reported performance gains, measured using LLM-based evaluation metrics on NLG datasets, translate to demonstrable improvements in generated output quality?
3) To further assess the method's capabilities, consider expanding the evaluation to include a wider range of generative tasks and datasets.

---

> ### Author Response · Authors · 2024-11-23
> **Response to HiNY (part 1)**
>
> Thanks for your valuable comments, we have conducted additional experiments to address the concerns on evaluation datasets, evaluation metrics, and hyper-parameters. The detailed responses to the questions and weaknesses are listed as follows.
>
> ### Response to Questions
>
> > 1. Given LoRA's established effectiveness in low-data fine-tuning scenarios, how does the proposed method perform under similar low-rank, data-constrained conditions?
>
> **Response:**  We have conducted additional experiments with low-rank settings ($r=32$) to investigate the effectiveness of GaLore $+$ under low-rank conditions. We have included two additional natural language generation (NLG) datasets (*i.e.*, CNN/DM and XSum), a foundation model (*i.e.*, LLaMA2-13B), and a state-of-the-art PEFT method (*i.e.*, DoRA [R-1]) as the baseline, in evaluation. The results on GSM8k, MAWPS, E2E, CNN/DM and XSum are listed in the following tables.
>
> *Table R1-1. Fine-tuning LLaMA2-7B and LLaMA2-13B on GSM8k with diverse ranks.*
>
> | Models     | Methods | Rank | Time(h)$\downarrow$ | Memory (GB) $\downarrow$ | Accuracy (%) $\uparrow$ |
> | ---------- | ------- | ---- | ---------------- | ------------------------ | ----------------------- |
> | LLaMA2-7B  | LoRA    | 32   | 0.53 | 15.65 | 23.30  |
> |            | DoRA    | 32   | 1.15 | **15.01** | 21.08  |
> |            | GaLore  | 32   | 3.48 | 15.42 | 26.46  |
> |            | GaLore $+$ | 32   | 0.88 | 15.23 | **29.11**  |
> |            | LoRA    | 128  | 0.53 | 16.99 | 30.78  |
> |            | DoRA    | 128  | 1.18 | 16.17 | 29.36  |
> |            | GaLore  | 128  | 3.50 | 16.06 | 33.66  |
> |            | GaLore $+$ | 128  | 0.92 | **15.73** | **34.65**  |
> |            | LoRA    | 256  | 0.55 | 18.79 | 33.36  |
> |            | DoRA    | 256  | 1.24 | 18.12 | 33.59  |
> |            | GaLore  | 256  | 3.53 | 16.66 | 31.92  |
> |            | GaLore $+$ | 256  | 0.92 | **15.74** | **33.81**  |
> | LLaMA2-13B | LoRA    | 32   | 0.73 | 27.96 | 32.01  |
> |            | DoRA    | 32   | 1.68 | **27.04** | 32.12  |
> |            | GaLore  | 32   | 7.08 | 28.14 | 36.24  |
> |            | GaLore $+$ | 32   | 1.22 | 27.93 | **37.98**  |
> |            | LoRA    | 128  | 0.77 | 30.11 | 33.99  |
> |            | DoRA    | 128  | 1.78 | 28.98 | 32.68  |
> |            | GaLore  | 128  | 7.20 | 29.28 | 40.03  |
> |            | GaLore $+$ | 128  | 1.37 | **28.90** | **41.24**  |
> |            | LoRA    | 256  | 0.80 | 33.08 | 34.72  |
> |            | DoRA    | 256  | 1.83 | 31.88 | 33.59  |
> |            | GaLore  | 256  | 7.25 | 30.06 | 37.15  |
> |            | GaLore $+$ | 256  | 1.54 | **29.79** | **42.20**  |
>
> *Table R1-2. Fine-tuning LLaMA2-7B and LLaMA2-13B on MAWPS with diverse ranks.*
>
> | Models     | Methods | Rank | Time(h) $\downarrow$ | Memory (GB) $\downarrow$ | Accuracy (%) $\uparrow$ |
> | ---------- | ------- | ---- | ---------------- | ------------------------ | ----------------------- |
> | LLaMA2-7B  | LoRA    | 32   | 0.40 | **14.36** | 45.80  |
> |            | DoRA    | 32   | 0.69 | 15.01 | 44.96  |
> |            | GaLore  | 32   | 2.59 | 15.15 | 58.40  |
> |            | GaLore $+$ | 32   | 0.62 | 14.91 | **60.50**  |
> |            | LoRA    | 128  | 0.45 | 15.64 | 65.97  |
> |            | DoRA    | 128  | 0.72 | 16.17 | 66.81  |
> |            | GaLore  | 128  | 2.61 | 15.79 | 64.29  |
> |            | GaLore $+$ | 128  | 0.62 | **15.44** | **67.64**  |
> |            | LoRA    | 256  | 0.40 | 17.60 | 61.76  |
> |            | DoRA    | 256  | 0.76 | 18.12 | 62.18  |
> |            | GaLore  | 256  | 2.66 | 16.39 | 63.03  |
> |            | GaLore $+$ | 256  | 0.69 | **16.12** | **65.55**  |
> | LLaMA2-13B | LoRA    | 32   | 0.50 | **26.50** | 55.04  |
> |            | DoRA    | 32   | 1.03 | 26.57 | 53.78  |
> |            | GaLore  | 32   | 5.34 | 27.94 | 61.34  |
> |            | GaLore $+$ | 32   | 0.80 | 27.59 | **63.87**  |
> |            | LoRA    | 128  | 0.53 | 28.77 | 54.62  |
> |            | DoRA    | 128  | 1.09 | 28.70 | 54.62  |
> |            | GaLore  | 128  | 5.39 | 28.99 | 65.13  |
> |            | GaLore $+$ | 128  | 0.91 | **28.54** | **69.75**  |
> |            | LoRA    | 256  | 0.55 | 31.62 | 62.18  |
> |            | DoRA    | 256  | 1.14 | 31.72 | 62.18  |
> |            | GaLore  | 256  | 5.50 | 29.87 | 62.18  |
> |            | GaLore $+$ | 256  | 1.06 | **29.45** | **66.39**  |

---

> > ### Author Response · Authors · 2024-11-23
> > **Response to HiNY (part 2)**
> >
> > *Table R1-3. Fine-tuning LLaMA2-7B and LLaMA2-13B on E2E with diverse ranks.*
> >
> > | Models     | Methods | Rank | Time(h) $\downarrow$ | Memory (GB) $\downarrow$ | BLEU $\uparrow$ | NIST $\uparrow$ | MET $\uparrow$ | ROUGE-1/2/L $\uparrow$ | CIDEr $\uparrow$ |
> > | ---------- | ------- | ---- | ---------------- | ---------------------- | --------------- | --------------- | -------------- | ---------------------- | ---------------- |
> > | LLaMA2-7B  | LoRA    | 32   | 2.94  | 14.17 | 62.47 | 4.58 | 35.07 | 69.16/38.00/46.24 | 1.46  |
> > |            | DoRA    | 32   | 5.14  | **14.13** | 62.63 | 4.59 | 35.11 | 69.25/38.07/46.31 | 1.47  |
> > |            | GaLore  | 32   | 18.57 | 15.15 | 64.95 | 4.96 | 36.92 | 70.99/40.77/49.15 | 1.77  |
> > |            | GaLore $+$ | 32   | 4.41  | 14.92 | **65.15** | **4.97** | **37.32** | **71.17**/**41.06**/**49.38** | **1.78**  |
> > |            | LoRA    | 128  | 3.02  | **15.43** | 64.60 | 4.86 | 36.74 | 70.67/40.20/48.54 | 1.70  |
> > |            | DoRA    | 128  | 5.27  | **15.43** | **64.77** | 4.86 | **36.77** | 70.72/40.20/48.62 | 1.72  |
> > |            | GaLore  | 128  | 18.45 | 15.79 | 64.22 | 5.14 | 36.50 | 70.99/41.19/49.37 | 1.80  |
> > |            | GaLore $+$ | 128  | 4.52  | **15.43** | 64.22 | **5.16** | 36.66 | **71.07**/**41.54**/**49.71** | **1.82**  |
> > |            | LoRA    | 256  | 3.01  | 17.43 | 64.93 | 4.94 | 36.81 | 70.92/40.66/49.01 | 1.76  |
> > |            | DoRA    | 256  | 5.61  | 17.67 | 64.94 | 4.95 | 36.81 | 70.94/40.72/49.04 | 1.77  |
> > |            | GaLore  | 256  | 18.42 | 16.39 | 64.97 | 5.10 | 36.84 | **71.19**/41.34/49.26 | 1.82  |
> > |            | GaLore $+$ | 256  | 4.86  | **16.19** | **64.99** | **5.11** | **36.88** | 71.15/**41.40**/**49.45** | **1.83**  |
> > | LLaMA2-13B | LoRA    | 32   | 3.63  | **26.31** | 62.00 | 4.69 | 34.52 | 68.98/38.31/46.62 | 1.52  |
> > |            | DoRA    | 32   | 7.55  | 26.44 | 61.99 | 4.70 | 34.58 | 69.00/38.40/46.65 | 1.51  |
> > |            | GaLore  | 32   | 38.40 | 27.94 | 65.06 | 4.90 | 37.00 | 70.90/40.68/48.97 | 1.76  |
> > |            | GaLore $+$ | 32   | 5.90  | 27.58 | **65.48** | **4.92** | **37.31** | **71.12**/**40.89**/**49.13** | **1.79**  |
> > |            | LoRA    | 128  | 3.65  | 28.54 | 64.99 | 4.82 | 36.97 | 70.70/40.18/48.41 | 1.72  |
> > |            | DoRA    | 128  | 8.08  | 28.69 | 64.87 | 4.81 | **36.99** | 70.70/40.18/48.36 | 1.70  |
> > |            | GaLore  | 128  | 38.12 | 28.99 | 64.35 | 5.05 | 36.48 | 70.81/41.04/49.17 | 1.78  |
> > |            | GaLore $+$ | 128  | 6.76  | **28.51** | **65.01** | **5.15** | 36.34 | **71.13**/**41.30**/**49.34** | **1.81**  |
> > |            | LoRA    | 256  | 3.89  | 35.97 | **65.27** | 4.89 | 37.12 | 70.89/40.60/48.79 | 1.74  |
> > |            | DoRA    | 256  | 8.48  | 31.60 | 65.12 | 4.88 | 37.09 | 70.92/40.45/48.55 | 1.72  |
> > |            | GaLore  | 256  | 38.30 | 29.87 | 65.13 | 5.00 | **37.15** | 71.19/41.17/49.10 | 1.80  |
> > |            | GaLore $+$ | 256  | 7.71  | **29.50** | 64.58 | **5.13** | 36.94 | **71.23**/**41.54**/**49.67** | **1.82**  |
> >
> >
> > *Table R1-4. Fine-tuning LLaMA2-7B and LLaMA2-13B on CNN/DM with rank=32.*
> >
> > | Models     | Methods | Time(h) $\downarrow$ | ROUGE-1 $\uparrow$ | ROUGE-2 $\uparrow$ | ROUGE-L $\uparrow$ |
> > | ---------- | ------- | ----------------- | ------------------ | ------------------ | ------------------ |
> > | LLaMA2-7B  | LoRA    | 0.37              | 31.68              | 11.45              | 21.85              |
> > |            | DoRA    | 0.68              | 31.98              | 11.57              | 22.01              |
> > |            | GaLore  | 1.18              | 33.64              | 12.98              | **24.27**          |
> > |            | GaLore $+$ | 0.70              | **33.76**          | **13.02**          | 23.40              |
> > | LLaMA2-13B | LoRA    | 0.51              | 32.77              | 12.14              | 22.98              |
> > |            | DoRA    | 0.94              | 32.69              | 12.00              | 23.10              |
> > |            | GaLore  | 2.23              | 33.65              | 13.06              | 24.33              |
> > |            | GaLore $+$ | 0.65              | **34.05**          | **13.25**          | **24.44**          |

---

> > > ### Author Response · Authors · 2024-11-23
> > > **Response to HiNY (part 3)**
> > >
> > > *Table R1-5. Fine-tuning LLaMA2-7B and LLaMA2-13B on XSum with rank=32.*
> > >
> > > | Models     | Methods | Time(h) $\downarrow$ | ROUGE-1 $\uparrow$ | ROUGE-2 $\uparrow$ | ROUGE-L $\uparrow$ |
> > > | ---------- | ------- | ----------------- | ------------------ | ------------------ | ------------------ |
> > > | LLaMA2-7B  | LoRA    | 0.25              | 37.25              | 13.46              | 29.50              |
> > > |            | DoRA    | 0.50              | 37.52              | 13.5               | 29.91              |
> > > |            | GaLore  | 1.04              | 40.24              | 16.07              | 32.92              |
> > > |            | GaLore $+$ | 0.36              | **40.36**          | **16.25**          | **33.00**          |
> > > | LLaMA2-13B | LoRA    | 0.35              | 39.15              | 15.11              | 31.53              |
> > > |            | DoRA    | 0.71              | 39.38              | 15.40              | 31.72              |
> > > |            | GaLore  | 2.03              | 42.54              | **18.08**          | 35.15              |
> > > |            | GaLore $+$ | 0.49              | **42.70**          | 18.06              | **35.26**          |
> > >
> > > As shown in Tables R1-1 to R1-5, GaLore $+$ consistently achieves comparable or superior results across various scenarios, with its advantages being particularly notable in Arithmetic Reasoning tasks. Experimental results show that the proposed method is specifically effective in low-data fine-tuning scenarios, which outperforms LoRA and other fine-tuning methods significantly.
> > >
> > > > 2. Do the reported performance gains, measured using LLM-based evaluation metrics on NLG datasets, translate to demonstrable improvements in generated output quality?
> > >
> > >   **Response:** We employ BLEU, NIST, METEOR, CIDEr, and ROUGE metrics to evaluate the performance on NLG. The meanings of the metrics are as follows.
> > >
> > > - BLEU [R-2] measures n-gram overlap between generated and reference texts, offering a way to evaluate lexical accuracy.
> > > - NIST [R-3] extends this by emphasizing the importance of informative n-grams, which aligns well with content-focused evaluations.
> > > - METEOR [R-4] incorporates stemming and synonym matching, making it effective for capturing semantic alignment in generative outputs.
> > > - ROUGE metrics (ROUGE-1, ROUGE-2, ROUGE-L) [R-5] are particularly popular for summarization tasks, as they assess content relevance through unigram, bigram, and sequence overlaps.
> > > - CIDEr [R-6] evaluates both consensus and diversity by comparing generated texts with multiple references using TF-IDF-weighted n-grams, making it applicable to tasks with varied yet valid outputs, such as captioning.
> > >
> > > All these methods are considered to enhance the quality of the generated contents. As far as we know, these metrics are widely used to evaluate the performance of PEFT methods on NLG tasks, such as, Table 3 in LoRA [R-7], Table 1 in Flora [R-8], Table 1 in NOLA [R-9].
> > >
> > > > 3. To further assess the method's capabilities, consider expanding the evaluation to include a wider range of generative tasks and datasets.
> > >
> > > **Response:** We have included two additional natural language generation (NLG) datasets (*i.e.*, CNN/DM and XSum) in evaluation. Besides, we have also included a foundation model (*i.e.*, LLaMA2-13B) and a new baseline method (*i.e.*, DoRA [R-1]) to thoroughly evaluate the capabilities of the proposed method. The detailed descriptions of the two additional NLG datasets are as follows.
> > >
> > > - CNN/DM: A dataset of news articles from CNN and Daily Mail with multi-sentence summaries, focusing on extractive and abstractive summarization tasks.
> > > - XSum: A dataset of BBC news articles with highly concise single-sentence summaries, emphasizing abstractive summarization.
> > >
> > > Different from the E2E dataset, the two datasets are designed for text summarization tasks. The results are listed in Tables R1-4 and R1-5.
> > >
> > > Experimental results show that the proposed method outperforms the baseline fine-tuning methods in most cases, indicating its effectiveness on generative tasks.
> > >
> > > ### Response to Weaknesses
> > >
> > > > 1. The evaluation datasets in experiment study are limited. Eval metrics for NLG(E2E) are not suitable for generative task evaluation.
> > >
> > > **Response:** Please refer to the responses to Question 2.
> > >
> > > > 2. The sensitivity of Hparams (rank, low data regime) is not well studied.
> > >
> > > **Response:** Please refer to the responses to Question 1.

---

> > > > ### Author Response · Authors · 2024-11-23
> > > > **Response to HiNY (part 4)**
> > > >
> > > > **References**:
> > > >
> > > > [R-1] Shih-Yang Liu, Chien-Yi Wang, Hongxu Yin, Pavlo Molchanov, Yu-Chiang Frank Wang, KwangTing Cheng, and Min-Hung Chen. Dora: Weight-decomposed low-rank adaptation. arXiv preprint arXiv:2402.09353, 2024.
> > > >
> > > > [R-2] Matt Post. A call for clarity in reporting bleu scores. arXiv preprint arXiv:1804.08771, 2018.
> > > >
> > > > [R-3] Ying Zhang, Stephan Vogel, and Alex Waibel. Interpreting bleu/nist scores: How much improvement do we need to have a better system? In LREC, 2004.
> > > >
> > > > [R-4] Tian-Tian Zou, Yu-Jie Zhou, Xiao-Dong Zhou, Wen-Yue Liu, Sven Van Poucke, Wen-Jun Wu, Ji-Na Zheng, Xue-Mei Gu, Dong-Chu Zhang, Ming-Hua Zheng, et al. Mets risk score: a clear scoring model to predict a 3-year risk for metabolic syndrome. Hormone and Metabolic Research, 50 (09):683–689, 2018.
> > > >
> > > > [R-5] Chin-Yew Lin. Rouge: A package for automatic evaluation of summaries. In Text summarization branches out, pp. 74–81, 2004.
> > > >
> > > > [R-6] Ramakrishna Vedantam, C Lawrence Zitnick, and Devi Parikh. Cider: Consensus-based image description evaluation. In Proceedings of the IEEE conference on computer vision and pattern recognition, pp. 4566–4575, 2015.
> > > >
> > > > [R-7] Edward J Hu, yelong shen, Phillip Wallis, Zeyuan Allen-Zhu, Yuanzhi Li, Shean Wang, Lu Wang, and Weizhu Chen. LoRA: Low-rank adaptation of large language models. In International Conference on Learning Representations, 2022. URL https://openreview.net/forum? id=nZeVKeeFYf9.
> > > >
> > > > [R-8] Yongchang Hao, Yanshuai Cao, and Lili Mou. Flora: Low-rank adapters are secretly gradient compressors. In Forty-first International Conference on Machine Learning, 2024. URL https: //arxiv.org/abs/2402.03293.
> > > >
> > > > [R-9] Soroush Abbasi Koohpayegani, Navaneet K L, Parsa Nooralinejad, Soheil Kolouri, and Hamed Pirsiavash. NOLA: Compressing loRA using linear combination of random basis. In The Twelfth International Conference on Learning Representations, 2024. URL https://openreview. net/forum?id=TjfXcDgvzk.

---

### Author Response · Authors · 2024-11-23
**General Response**

We sincerely thank all the reviewers for their valuable feedback to improve the quality of our manuscript. We have updated the manuscript with the revised contents highlighted in blue. The main changes are summarized as below.

**Low-rank projections in GaLore $+$**

- During the rebuttal period, we found the mistake of Figure 2b. We previously mistakenly calculated separate projection matrices for each attention head and used these matrices to compute the approximation error for each head. Thus, the approximation error is lower than that of exact SVD. We have addressed this mistake and revised the manuscript. Moreover, we have carefully checked our codes for the main experimental results and there were no mistakes.

- We have revised the descriptions on the low-rank projection to improve the presentation.

**Extended experiments**

- We have conducted additional experiments with low-rank settings ($r=32$) to investigate the effectiveness of GaLore $+$ under low-rank conditions.
- We have included two additional natural language generation (NLG) datasets (*i.e.*, CNN/DM and XSum), two foundation models (*i.e.*, LLaMA2-13B, LLaMA3-8B), and a state-of-the-art PEFT method (*i.e.*, DoRA) as the baseline, in evaluation.
- We have included an ablation study on the sparsely coded residual with multiple random seeds for convincing results.

---

### Meta-Review · Area_Chair_ZkAS · 2024-12-14

**Metareview:**

The work aims to improve Low-Rank Adaptation for LLMs by using cross-head low-rank projection, and a more efficient alternative to SVD.
Unfortunately, several concerns remained about the paper. First, on the statistical significance and rigor of the experimental results, including only relatively high ranks instead of lower ranks often used in practice, performance improvements often fell within one standard deviations, and there was unclarity if the sparsely coded residual component actually contributed meaningful improvements. Second, we found it inadequately explained how GaLore+ could outperform vanilla GaLore in the cases without sparsely coded residuals, where the methods would then coincide.

We hope the detailed feedback helps to strengthen the paper for a future occasion.

**Additional Comments On Reviewer Discussion:**

The author feedback phase was useful as acknowledged by the reviewers. Some of the concerns however remained if the work is ready for the high bar of ICLR.

---

### Decision · Program_Chairs · 2025-01-22

Reject